# Climate change and elevated $CO_2$ favor forest over savanna under different future scenarios in South Asia

Dushyant Kumar[1], Mirjam Pfeiffer[1], Camille Gaillard[1], Liam Langan[1], and Simon Scheiter[1]

[1]Senckenberg Biodiversity and Climate Research Centre (SBiK-F), Senckenberganlage 25, 60325 Frankfurt am Main, Germany

**Correspondence:** Dushyant Kumar (dushyant.kumar@senckenberg.de)

**Abstract.**

South Asian vegetation provides essential ecosystem services to the 1.7 billion inhabitants living in the region. However, biodiversity and ecosystem services are threatened by climate and land-use change. Understanding and assessing how ecosystems respond to simultaneous increases in atmospheric $CO_2$ and future climate change is of vital importance to avoid undesired
ecosystem change. Failed reaction to increasing $CO_2$ and climate change will likely have severe consequences for biodiversity and humankind. Here, we used the aDGVM2 to simulate vegetation dynamics in South Asia under RCP4.5 and RCP8.5, and we explored how the presence or absence of $CO_2$ fertilization influences vegetation responses to climate change. Simulated vegetation under both RCPs without $CO_2$ fertilization effects showed decrease in tree dominance and biomass, whereas simulations with $CO_2$ fertilization showed an increase in biomass, canopy cover, and tree height and a decrease in biome-specific
evapotranspiration by the end of the 21st century. The predicted changes in above ground biomass and canopy cover triggered transition towards tree-dominated biomes. We found that savanna regions are at high risk of woody encroachment and transitioning into forest. We also found transitions of deciduous forest to evergreen forest in the mountain regions. Vegetation types using $C_3$ photosynthetic pathway were not saturated at current $CO_2$ concentrations and the model simulated a strong $CO_2$ fertilization effect with the rising $CO_2$. Hence, vegetation in the region has the potential to remain a carbon sink. Projections
showed that the bioclimatic envelopes of biomes need adjustments to account for shifts caused by climate change and elevated $CO_2$. The results of our study help to understand the regional climate-vegetation interactions and can support the development of regional strategies to preserve ecosystem services and biodiversity under elevated $CO_2$ and climate change.

## 1   Introduction

Global climate has been identified as the primary determinant of large-scale natural vegetation patterns (Overpeck et al., 1990). Climate change has affected global vegetation pattern in the past and caused numerous shifts in plant species distribution over the last few decades (Chen et al., 2011; Thuiller et al., 2008). It is expected to have even more pronounced effects in the future

and may lead to drastically increasing species extinction rates in various ecosystems (Brodie et al., 2014). Natural ecosystems have been and continue to be exposed to increased climate variability and abrupt changes caused by increased intensity and frequency of extreme events such as heat waves, drought and flooding (Herring et al., 2018). At the same time, they are under severe pressure due to anthropogenic disturbance and land conversion. Rising levels of atmospheric $CO_2$ are a strong driver of climate-induced vegetation changes (Allen et al., 2014). Anthropogenic $CO_2$ emissions account for approximately 66% of the total anthropogenic greenhouse forcing (Forster et al., 2007) and are thus largely responsible for contemporary and future global climate change Parry et al. (2007). Rising $CO_2$ is expected to alter distributions of plant species and ecosystems (Parry et al., 2007) both indirectly through its influence on global temperatures and precipitation patterns (Cao et al., 2010), two main drivers of vegetation dynamics, and directly via its physiological effects on plants (Nolan et al., 2018). It is therefore of vital importance to understand how ecosystems respond to simultaneous increases in atmospheric $CO_2$ and temperature, changes in precipitation regime, and to altered ecosystem water balance in order to avoid critical ecosystem disruptions and the resulting consequences for biodiversity and humankind.

Increases in temperatures, decreases in precipitation as well as changes in precipitation seasonality can cause loss of vegetation biomass. Plant using $C_3$ photosynthetic pathways are often not saturated at the current atmospheric $CO_2$, whereas plants using the $C_4$ photosynthetic pathways are already at their physical optimum at current atmospheric $CO_2$ levels (Ehleringer and Cerling, 2002). The physiology of $C_3$ plants implies that elevated atmospheric $CO_2$ improves their ability for carbon uptake due to the $CO_2$ fertilization (Woodrow and Berry, 1988) and enhances carbon sequestration (Leakey et al., 2009; Norby and Zak, 2011) as well as plant water use efficiency (Soh et al., 2019). This has also been observed in Long-term Long-term Free-Air Carbon dioxide Enrichment (FACE) experiments (Norby and Zak, 2011). Thus, elevated $CO_2$ influences photosynthesis and thereby affects other physiological processes such as respiration, decomposition (Doherty et al., 2010), evapotranspiration (ET) and biomass accumulation (Frank et al., 2015). Increasing $CO_2$ concentration has been associated with woody cover increase in structurally open tropical biomes such as grasslands and savannas (Stevens et al., 2017). This widespread proliferation of woody plants into arid and semiarid ecosystems has been attributed to increased water use efficiency in $C_3$ plants that facilitates woody sapling establishment and growth due to higher drought tolerance (Kgope et al., 2010; Stevens et al., 2017). These $CO_2$ effects on plant growth and competition can alter community structure (height distribution), ecosystem productivity, climatic niches of ecosystems and biome boundaries (Nolan et al., 2018; Wingfield, 2013).Change in vegetation distribution and altered vegetation structure feed back on climate by altering fluxes of energy, moisture and $CO_2$ between land and atmosphere (Friedlingstein et al., 2006). Feedback mechanisms also involve vegetation-mediated changes in albedo, surface roughness, land-atmosphere fluxes and evapotranspiration, (Field et al., 2007; Richardson et al., 2013).

Enhanced plant growth due rising $CO_2$ implies rapid leaf area development and more total leaf area could translate into higher transpiration (Leakey et al., 2009). However, elevated $CO_2$ concentrations may decrease leaf stomatal conductance to water vapor which could reduce transpiration. Evapotranspiration (ET) is a key ecophysiological process in the soil-vegetation-atmosphere continuum (Feng et al., 2017). Annually, 64% of the total global land-based precipitation is returned to the atmosphere through ET (Zhang et al., 2016). Environmental change and concurrent vegetation changes alter ET and affect water

availability (Mao et al., 2015), especially in arid and semiarid regions. In these regions, ET affects surface and subsurface processes such as cloud development, land surface temperature, and groundwater recharge (Fisher et al., 2011).

South Asia is home to approx. 1.7 billion people and is one of the regions most vulnerable to climate change (Eckstein et al., 2018). It hosts four of the world's biodiversity hotspots (Myers et al., 2000) and harbours different biome types ranging from tropical in the south to temperate in the north at the fringe of the Himalayas. These hotspots are characterized by high levels of diversity and endemism, and they are threatened by climate change and anthropogenic land-use (Deb et al., 2017). For instance, woody encroachment due to rising $CO_2$ threatens South Asian savannas (Kumar et al., 2020) and sifting cultivation in the north eastern part of South Asia threatens biodiversity (Bera et al., 2006).

Due to the absence of long-term field experiments such as FACE experiments, in the dominant biomes of the region, modeling studies are valuable tools to close existing knowledge gaps. Dynamic global vegetation models (DGVMs, Prentice et al., 2007) are particularly well-suited to address questions that focus on vegetation response to changing environmental drivers, e.g., climate and $CO_2$. While most DGVM studies in South Asia focused on vulnerability of forests to climate change (Chaturvedi et al., 2011; Ravindranath et al., 2006, 1997) they often overlooked the severely threatened savanna biome. These studies were further limited by the utilization of models with fixed eco-physiological parameters and traits e.g., fixed carbon allocation values to assign carbon to plant biomass pools, fixed specific leaf area (SLA), as well as pre-defined bioclimatic limits that were derived from contemporary climatology in order to constrain the spatial distribution of plant functional types (PFTs). Moreover, many DGVMs used in these studies do not account for life history, eco-evolutionary processes and trait variability among individual plants (Kumar and Scheiter, 2019). While some global-scale studies have investigated the potential effect of increasing $CO_2$ on natural vegetation, carbon sequestration and biome boundaries (e.g., Hickler et al., 2006; Sato et al., 2007; Smith et al., 2013), detailed modeling studies focusing explicitly on different biomes in South Asia have not been conducted. The physiological effects of increased $CO_2$ and climate change on South Asian vegetation is uncertain and needs to be addressed in order to improve understanding of regional ecosystem functioning as well as implications for biodiversity conservation.

To address the knowledge gaps in existing studies, we used the aDGVM2 (adaptive dynamic global vegetation model version 2), an individual- and trait-based vegetation model that combines elements of traditional DGVMs (Prentice et al., 2007) with newly implemented approaches for selection and trait filtering. In aDGVM2, environmental conditions select for the plants with trait value combination that make them successful under these conditions. Therefore, plant communities that are adapted to site-specific environmental conditions dynamically assembles and emerge as a reaction the environmental forcing (Langan et al., 2017; Scheiter et al., 2013). Originally, aDGVM2 had been tested for Amazonia Langan et al. (2017) and Africa (Gaillard et al., 2018; Pfeiffer et al., 2019). In order to adapt it to South Asian ecosystems and their diversity, we included $C_3$ grasses, improved ecophysiological processes such as the leaf energy budget in order to estimate leaf temperature, implemented separate temperature sensitivities for $C_3$ and $C_4$ photosynthetic capacity ($V_{cmax}$ and included snow in the water balance model.

In this study we used the updated version of aDGVM2 and addressed the following questions:

1. How do projected changes in climate and $CO_2$ following two Representative Concentration Pathways (RCP8.5 and RCP4.5, Meinshausen et al., 2011) change the distribution, boundaries and climatic niches of biomes in South Asia?

2. How does the relationship between projected biomass, ET, temperature and precipitation change in response to $CO_2$ fertilization?

3. What is the sensitivity of predicted changes in relation to presence and absence of $CO_2$ fertilization?

Based on our results we analyzed climate-vegetation interactions to improve our understanding of how to manage and mitigate impacts on biomes under climate change and increasing $CO_2$.

## 2 Methods

### 2.1 Description of the study region

Approx. 1.7 billion people populate South Asia, i.e., the Indian subcontinent, Afghanistan and Myanmar. South Asia incorporates a wide range of bio-climatic zones with distinctive biomes, ecosystem types and species (Rodgers and Panwar, 1988). Climatic conditions are controlled by interactions between the South Asian summer monsoon system and the region's complex topography. The climatic envelope ranges from tropical arid and semi-arid regions in the west, to humid tropical regions supporting rainforests in the northeast and temperate vegetation at the fringe of the Himalaya. Excluding the Himalayan regions, South Asia has a mean annual temperature of approximately 24°C with very low spatial variability. Mean annual precipitation (MAP) is 1190 mm, ranging from less than 500 mm in the warm desert zone in the west to more than 3500 mm in the northeast. The steep elevation gradients ranging from sea level to 8800 m result in a rich diversity of ecosystems that can alternate in areas of a few hundred square-kilometres. Topography is recognized as a strong driver of ecological patterns, for example those related to forest structure and composition, floristic diversity, and soil fertility (Gallardo-Cruz et al., 2009; Jucker et al., 2018; Sinha et al., 2018). South Asia hosts four major global biodiversity hotspots, namely the Western Ghats, Himalayas, Indo-Myanmar and Sri Lanka (www.conservation.org, Conservation International, 2013, Myers et al., 2000). These hotspots include a wide diversity of ecosystems such as mixed wet evergreen, dry evergreen, deciduous, and montane forests. Further vegetation types are alluvial grasslands and subtropical broadleaf forests along the foothills of the Himalayas, temperate broadleaf forests in the mid hills, mixed conifer and conifer forests in the higher hills, savanna in the Deccan region and southern part of Malaysia, and alpine meadows above the tree line (Conservation International, 2013).

### 2.2 Model Description

For this study we used aDGVM2 (Scheiter et al., 2013; Langan et al., 2017; Gaillard et al., 2018), a DGVM with a dynamic trait approach. In the supplementary material we summarize main features of aDGVM2 and explain how the physiological effect of changing $CO_2$ concentration and rising temperature are simulated in a process based way in the aDGVM2 by implemented photosynthesis routine. To adapt the aDGVM2 to the requirements of the study region, we incorporated new sub-routines into the model. We improved the representation of (a) the water balance by including snow, (b) the carboxylation rate, (c) leaf temperature, and (d) we included $C_3$ grasses (previous model versions only simulated $C_4$ grasses).

(a) Water balance. In aDGVM2, the soil water module is based on the tipping-bucket concept. As the model was originally developed with strong focus on tropical and subtropical forest and savanna regions, the original model version only considered water input in form of rain (see Langan et al., 2017). In the updated model version, precipitation is assigned as snow when daily mean air temperature drops below 0°C. Snow accumulates on the soil surface or is added to the top of an existing snowpack. The snowpack persists as long as air temperature remains below 0°C. Once temperature rises above 0°C, water from snowmelt is added to the soil water pool and becomes available to plants. This process may improve the water availability for plants at the beginning of spring, for example in the Himalayan region. Snowmelt ($S_{melt}$, mm/day) is calculated following (Choudhury et al., 1998) as

$$S_{melt} = 1.5 + K_m P_{precip}(T_a - T_{snow})S_{pack}, \tag{1}$$

where $K_m$ is the coefficient of snowmelt (0.007 mm/day/ °C), $S_{pack}$ is the depth of the snowpack (mm) and is equialent to the accumulated soilid portion of precipitation, $T_a$ is daily mean air temperature (°C), $P_{precip}$ is precipitation (mm/day) and $T_{snow}$ is the maximum temperature where precipitation falls as snow (0°C). We do not consider insulation effects of the snowpack in the model.

(b) Carboxylation rate.

In earlier versions of aDGVM2, leaf-level photosynthesis was calculated at population level, i.e., it was assumed that all plants of a simulated vegetation stand have the same leaf-level photosynthetic rate. Only $C_3$- and $C_4$-type photosynthesis were distinguished. We therefore implemented new routines to calculate photosynthesis at a daily time step for each individual plant. We further incorporated an empirical relation between specific leaf area ($A_{SLA}$, mm$^2$/mg) and leaf nitrogen content per unit area ($N_a$, g/m$^2$) following Sakschewski et al. (2015),

$$N_a = 6.89 A_{SLA}^{-0.571}, \tag{2}$$

The standard maximum carboxylation rate of rubisco per leaf area ($V_{cmax,25}$, $\mu$mol/m$^2$/s) was derived from the TRY database Kattge and Knorr (2007) by Sakschewski et al. (2015) and is calculated as

$$V_{cmax,25} = 31.62 N_a^{0.501}, \tag{3}$$

where $V_{cmax,25}$ is $V_{cmax}$ at 25°C.

In the model, $A_{SLA}$ is linked to the matric potential at 50% loss of xylem conductance (P50, see Langan et al., 2017). The trade-off between $A_{SLA}$ and $V_{cmax}$ mediated by leaf traits ($N_a$) introduces variability in the spectrum of tree growth strategies in aDGVM2. In addition to the $A_{SLA}$ is linked to leaf longevity (LL) in aDGVM2, such that it affects the leaf turnover rates (represented by Equation 72, in Appendix, Langan et al (2017)). Leaves with high $A_{SLA}$ have shorter LL and higher turnover rates than leaves with low $A_{SLA}$ (and vice-versa). The correlation between $A_{SLA}$, P50 and LL represent the trade-off between two opposing resource strategies, i.e., conservation vs. rapid acquisition of soil water and nutrients (Wright et al., 2005). Trees that invest more carbon into their (low $A_{SLA}$) enhances their structural stability, but have lower leaf turnover to mitigate the higher initial carbon investment.

The effect of temperature on photosynthesis is well-described (Kirschbaum, 2004), and temperature may influence photosynthesis both directly, via temperature-dependency of enzyme-mediated metabolic rates of carboxylation and the Calvin cycle (Sharkey et al., 2007), and indirectly via its effect on transpiration and plant water uptake and transport (Urban et al., 2017). The maximum carboxylation rate ($V_{cmax}$) increases with temperature until it reaches an optimum, and decreases again at temperatures above the optimum (Kattge and Knorr, 2007) due to reductions in enzyme activity. Above 30°C the electron

transport chain is gradually inhibited, and at temperatures above 40°C the denaturation of Rubisco and associated proteins becomes relevant (Lloyd et al., 2008). The temperature dependency of the carboxylation rate ($V_{cmax}$) is expressed as

$$V_{cmax} = \frac{V_{cmax,25} 2^{0.1(T_{leaf}-25)}}{(1+e^{0.3(T_{low}-T_{leaf})})(1+e^{0.3(T_{leaf}-T_{upp})})}, \tag{4}$$

where $T_{leaf}$ is the leaf temperature in °C (see next paragraph for calculation). The photosynthetic model of Collatz et al. (1992) and Collatz et al. (1992) assumes specific values of $T_{upp}$ and $T_{low}$ for C$_3$ and C$_4$ plants, respectively (Table S1 and Table

S2). These temperature ranges from -10°C to 36°C and 13°C to 45°C for C$_3$ and C$_4$ photosynthetic pathways respectively, allow plants to grow most efficiently in their plant-specific climatic niches.

   (c) Leaf temperature. We calculate leaf temperature following the leaf-level energy budget concept (Gates, 1968). Leaf-level photosynthesis, activity of leaf enzymes and transpiration depend on leaf temperature ($T_{leaf}$,°C), calculated as

$T_{leaf} = T_{air} + (\frac{R_n - \lambda E r_{gb}}{\rho C_P}), \tag{5}$

where $T_{air}$ is air temperature (°C), $R_n$ is net radiation absorbed by the leaf (MJ/m$^2$/day), $\lambda$ is latent heat of vaporization (MJ/kg), E is evapotranspiration (m/day), $r_{gb}$ is the boundary layer resistance (m/s), $\rho$ is the air density (kg/m$^3$) and derived from atmospheric pressure (101.325 kPa at sea level) that is scaled according to the elevation and $T_{air}$, and $C_P$ is the specific heat of dry air (MJ/kg/°C). Leaf temperature is used to calculate the temperature dependence of $V_{cmax}$ used in the photosynthesis model

routines in equation (4). Absorbed net radiation ($R_n$), $r_{gb}$ and E are model state variables calculated from climate input used in aDGVM2 ($T_{air}$, long-wave and short-wave radiation) and $\rho$ is derived from atmospheric pressure (101.325 kPa at sea level) that is scaled according to the elevation and $T_{air}$. The value of latent heat of vaporization ($\lambda$), and $C_P$ are 2.45 MJ/kg and 2.71 MJ/kg/°C respectively, and are assumed as constant parameters in this model version.

(d) C$_3$ grasses. C$_3$ grasses were not included in previous aDGVM2 versions (Gaillard et al., 2018; Langan et al., 2017; Pfeiffer et al., 2019; Scheiter et al., 2013). We therefore implemented C$_3$ grasses, following the approach used for C$_4$ grasses in previous model versions but adjusted the photosynthetic pathway (see Appendix S2 in Langan et al., 2017). C$_3$ and C$_4$ grasses use a different leaf-level photosynthesis model (Farquhar et al., 1980) following the implementations of Collatz et al. (1991, 1992). The optimum temperature ranges for carboxylation for C$_3$ and C$_4$ grasses are also different (Table S1). As C$_3$ grasses

have higher cold tolerance than C$_4$ grasses (Liu and Osborne, 2008), we implemented frost intolerance for C$_4$ grasses but not for C$_3$ grasses. Frost is assumed to damage the tissue of C$_4$ grasses, and in aDGVM2 we kill 10% of the living leaf biomass of C$_4$ grasses per frost day independent of frost severity.

### 2.3 Model forcing data

#### 2.3.1 Climate data

We used GFDL-ESM2M climate data for the period 1950 to 2099 from the Inter-Sectoral Impact Model Inter-comparison Project (ISIMIP2), as historical climate simulated by GFDL-ESM2M showed satisfactory performance for South Asia (McSweeney and Jones, 2016). The general circulation model (GCM) output was bias-corrected in ISIMIP and downscaled to a spatial resolution of $0.5° \times 0.5°$ (Warszawski et al., 2014). We used average, maximum and minimum air temperatures, precipitation, surface downwelling shortwave radiation and long-wave radiation, near-surface wind speed, and relative humidity at a daily temporal resolution. We used two representative concentration pathways, namely RCP4.5 and RCP8.5 (Meinshausen et al., 2011). These scenarios assume increases in radiative forcing of 4.5 and 8.5 Wm$^2$ by 2100 (Van Vuuren et al., 2011) and increases of atmospheric $CO_2$ concentrations to 560 ppm and 970 ppm by 2100, respectively (Van Vuuren et al., 2011).

#### 2.3.2 Projected changes in temperature and precipitation

Mean annual precipitation (MAP) from GFDL-ESM2M does not show a clear trend when averaged for South Asia under RCP4.5 and RCP8.5, due to high inter-annual variability of precipitation (Fig. S1). Yet, there are region-specific differences in precipitation change. The Western Ghats which located between 73°- 77' E and 8°N - 21°N and eastern Himalayan region are projected to become wetter under both RCP4.5 and RCP8.5, whereas the western part of the region is projected to become drier by the end of the century under both RCPs (Fig. S2). MAP is projected to increase by more than 600 mm in the Eastern Himalayas and Western Ghats, but predicted to decrease by 400-600mm in the western and central area of the region (Fig. S2). By the end of the 21st century, mean annual temperature (MAT) of South Asia is expected to increase between ca. 1°C and 3.5°C under RCP4.5 and between 1°C and 6°C under RCP8.5, relative to the average temperature in the baseline period of 2000–2009 (Fig. S1 and Fig. S2). The western parts of the region and the Himalayan mountains are projected to experience higher increases in temperature than the rest of the region (Fig. S2).

#### 2.3.3 Soil and elevation data

Soil data was obtained from FAO (http://www.fao.org/soils-portal, Nachtergaele et al., 2009) and includes information on soil properties and types. The soil properties include parameters required by aDGVM2: volumetric water-holding capacity, soil hydraulic conductivity, soil bulk density, soil depth, soil texture, soil carbon content, soil wilting point and field capacity (for details see Fig. S3b and Langan et al., 2017). A digital elevation model (DEM) at 90m spatial resolution was obtained from the Shuttle Radar Topography Mission (SRTM, Fig. S3a http://srtm.csi.cgiar.org, Jarvis et al., 2008). It was resampled to a spatial resolution of $0.5° \times 0.5°$, to match the spatial resolution of climate data. In the model, elevation is used to calculate the surface pressure at a given altitude, which is used to scale up air density and partial pressure of oxygen. The partial pressure of oxygen is used to estimate the $CO_2$ compensation point of photosynthesis (Eq. 2 of Appendix S2 in Langan et al., 2017). We did not use slope and aspect in the model.

## 2.4 Model simulation protocol

To understand how climate and $CO_2$ fertilization interact to influence the future vegetation state in South Asia, we simulated all combination of two climate scenarios (RCP4.5 and RCP8.5) and two $CO_2$ scenarios ($CO_2$ fertilization enabled or disabled, four scenarios in total). We simulated potential natural vegetation between 1950 and 2099 using daily climate data for RCP4.5 and RCP8.5 (see section 2.3.1). For both scenarios, simulations were run with $CO_2$ increase in line with RCP4.5 (hereafter RCP4.5+e$CO_2$) and RCP8.5 (hereafter RCP8.5+e$CO_2$) and with the same climate data but fixed $CO_2$ after 2005 at 375 ppm for RCP4.5 (hereafter RCP4.5+f$CO_2$) and RCP8.5 (hereafter RCP8.5+f$CO_2$). Fixing the $CO_2$ concentration after 2005 mimics a situation where $CO_2$ fertilization would not occur and vegetation only responds to the climate signal. All simulations were conducted with natural fire as implemented in aDGVM2 and at 0.5°× 0.5°spatial resolution. The aDGVM2 simulates 1 hectare stands that are assumed to be representative for the vegetation at larger scale, i.e., we assume that the stand-level vegetation homogeneously covers all hectares within a simulated grid cell. The "representative hectare approach" is a concession to computational limitation, as photosynthesis and physiological processes are simulated individually for all individual plants of a stand ( upto 36000 individuals). It balances adequate representation of trait diversity among individual against computational constraint. Also due to computation time constraints, we did not conduct replicate simulations.

To ensure that simulated vegetation had sufficient time to adapt to prevailing environmental conditions, we conducted simulations for 650 years, split into a 500 year spin-up phase and a 150 year transient phase. For the spin-up phase, we randomly sampled years of the first 30 years of daily climate data (1950 to 1979). For the transient phase, we used the sequence of daily climate data between 1950 and 2099. Trial simulations showed that a 500 year spin-up period is sufficient to ensure that vegetation is in a dynamic equilibrium state with environmental drivers.

## 2.5 Model benchmarking and evaluation

For benchmarking of aDGVM2 simulation results, we used five different remote sensing products: aboveground biomass (t/ha, Saatchi et al., 2011), tree height (m, Simard et al., 2011), tree cover (percent, Friedl et al., 2010), MODIS evapotranspiration (mm/year, Zhang et al., 2010) and natural vegetation type (Ramankutty et al., 2010). We used a 10 year average of MODIS ET and compared it to a 10 year average of model simulated ET (mm/year; 2000-2009). All remote sensing data sets were aggregated to a 0.5° × 0.5° spatial resolution, to match the spatial resolution of model simulations by calculating the mean of all values within each 0.5° grid cell, or using nearest neighbour aggregation in the case of vegetation type ("raster" package in R, Hijmans and van Etten, 2012). We first compared model results and observations assuming that the entire study region is covered by natural vegetation (Figs.1). Then we repeated the comparisons only for areas with predominantly natural cover, i.e., we masked out areas with more than 50% managed land (Figs. S4, land cover classes 7 'Cultivated and Managed Vegetation' and 9 'Urban and Built-up' in, Tuanmu and Jetz, 2014). We calculated Normalized Mean Squared Error (NMSE) and coefficient of determination ($R^2$ to quantify agreement between data and simulated variables.

## 2.6 Biome classification

The aDGVM2 simulates state variables such as biomass and canopy cover of individual plants in simulated vegetation stands (1 hectare which is a representative of grid cell). We used woody canopy area, abundance of shrubs and trees, and grass biomass to classify the simulated vegetation into biome types (Fig. S5). We used 10-year averages of state variables for the periods 2000-2009, 2050-2059 and 2090-2099 to represent the 2000s, 2050s and 2090s, respectively. We classified areas with woody canopy cover below 5% as barren if grass biomass was below 100 kg/ha, and as grassland if grass biomass exceeded 100kg/ha. Grassland was classified as $C_3$ grassland or $C_4$ grassland based on predominance of $C_3$ or $C_4$ grass biomass. Simulated woody individuals were classified as trees if they had three or less stems and as shrubs if they had four or more stems (see supplementary material). The canopy cover of woody plants and grass biomass were used to separate woodland and savanna biomes. Grid cells with tree canopy cover greater than shrub canopy cover, tree canopy cover between 5% and 45%, and grass biomass below 100 kg/ha, were classified as woodland. Grid cells with the same woody cover characteristics but grass biomass higher than 100kg/ha were classified as savanna. Savanna was further separated into $C_4$ savanna or $C_3$ savanna based on the predominance of $C_3$ or $C_4$ grass biomass. Areas with canopy cover greater than 45% were classified as forest if tree cover exceeded shrub cover, or shrubland if shrub cover exceeded tree cover, irrespective of grass biomass. Forests were subdivided into evergreen and deciduous forest based on the dominance of canopy area of both tree phenology types. In aDGVM2, whether a plant is deciduous or evergreen is decided by a trait that is used in the categorization of plant community. Biomes considered in this study were hence $C_3$ grassland, $C_4$ grassland, shrubland, woodland, deciduous forest, evergreen forest, $C_3$ savanna and $C_4$ savanna.

Biomes differ in the amount of precipitation they receive and their temperatures. Whittaker plots describe the boundaries of observed biomes with respect to temperature and precipitation. We used the "plotbiomes" R-package (https://github.com/valentinitnelav/plotbiomes by Valentin Ștefan) to create Whittaker plots based on Ricklefs (2008) and (Whittaker, 1978). We overlaid the simulated biomes on Whittaker plots to assess at climatic niches of biomes under current climate to determine shifts in climatic niches by the end of this century as a result of climate change and elevated $CO_2$ under both RCPs (see section 3.6).

## 2.7 Calculation of biome-level evapotranspiration

For analyzing evapotranspiration change we calculated the amount of water transpired per unit leaf biomass. Simulated ET and leaf biomass for woody plants, $C_3$ grass and $C_4$ grass were summed and scaled to the grid level, taking latitudinal variation of grid cell area into account. Absolute change in evapotranspiration quantity can either result from the change in biome area or from a change in total amount of leaf-biomass over time or from changes in water use efficiency. In order to eliminate the effects caused by change in biome area and leaf biomass, we calculated biome level evapotranspiration by normalizing evapotranspiration with biome-level leaf biomass (Equation 6). Due to the normalization differences in evapotranspiration at biome level, are comparable between different biomes and independent from biome attributes such as its spatial extent and biome-level biomass. The biome-level evapotranspiration is calculated as the ratio of total annual ET over total leaf biomass

for all respective biomes:

$$E_{biome} = \frac{\sum_{i=1}^{G}(E_{grid,i}A_{grid,i})}{\sum_{i=1}^{G}(B_{grid,i}A_{grid,i})} \tag{6}$$

where $E_{biome}$ is biome-level ET (mm/kg/year), 1, 2, …, G represent the grid cells of the biome, $A_{grid,i}$ is the area of grid cell i (m$^2$, $E_{grid,i}$ is total evapotranspiration of grid cell i (mm/year), $B_{grid,i}$ is leaf biomass of grid cell i (kg/m$^2$. Choosing to normalize evapotranspiration to leaf biomass integrates over both increased water use efficiency and soil water availability constraints. It is therefore suitable to characterize overall change in the water balance over time at biome level, as it not only indicates water used to produce new biomass (as GPP over transpiration would express), but also includes water required to sustain existing biomass. We calculated the percentage change in $E_{biome}$ for respective scenarios between the 2010s and 2050s, and between the 2010s and the 2090s.

## 3  Results

### 3.1  Model performance and contemporary vegetation patterns

The aDGVM2 captured contemporary large-scale patterns of biomass, canopy cover, tree height and evapotranspiration. Model results agreed well with remote sensing products used for benchmarking (Fig.1). R$^2$ was 0.61, 0.45, 0.6 and 0.71, and NMSE was 0.48, 0.78, 0.4 and 1.07 for biomass (Saatchi et al., 2011), tree height (Simard et al., 2011), tree cover (Friedl et al., 2010) and evapotranspiration (Zhang et al., 2010), respectively (Figs.1 and 2). Data-model agreement improved when masking out managed land (Tuanmu and Jetz, 2014). R$^2$ increased to 0.66, 0.71, 0.67 and 0.80, while NMSE decreased to 0.43, 0.30, 0.61 and 1.03 for biomass, tree height, tree cover and evapotranspiration, respectively (Fig.S4). The model performed well in areas with higher fractional cover of natural vegetation, such as the Himalayas, Western Ghats and the northeast of the region, although the model overestimated biomass and canopy area in the Brahmaputra basin which lies between 28°N - 34°N and 90°E - 96.5°E in the northeast of the study region (Fig.1a,c, Kumar et al., 2020).

The model simulated evergreen forests along the Himalayan mountains, southern part of the Western Ghats and Sri Lanka, whereas deciduous forest was simulated in the northern Western Ghats, central India and southern parts of Myanmar (Fig.2a). Savanna was simulated in southern, northern and western parts of India and some regions of central Myanmar. Shrublands were simulated in the arid regions of Pakistan, the western parts of India and Afghanistan. The aDGVM2 simulated woodland in the west of central India, and grassland in the drier regions (Fig.2a). A large proportion of simulated deciduous forest area is in good agreement with that in maps of potential natural vegetation (PNV, Figs.2b,c). However a large proportion of simulated savanna area is represented as deciduous forest in the map of PNV (Fig.2b).

### 3.2  Projected changes in biome distribution pattern

The aDGVM2 projected increasing trends for canopy cover and above ground biomass in response to climate change and CO$_2$, and hence, changes in biome type, predominantly from savanna and grassland to deciduous forest (Fig.3a,b). Simulations

showed an increase in the area covered by evergreen and deciduous forests under both scenarios with $eCO_2$, in contrast to simulations under both scenarios with $fCO_2$ where $CO_2$ was fixed after 2005 (Table.1). Under RCP4.5+$eCO_2$, evergreen and

deciduous forest cover increased by 3.1% and 21.2% until the 2050s, and 38.0% and 59.1% until the 2090s, respectively. Under RCP8.5+$eCO_2$, evergreen and deciduous forest increased by 24.8% and 45.4% until the 2050s, and 46.5% and 60.2% until the 2090s, respectively. The model simulated a small increase in forest area for RCP4.5+$fCO_2$, where the area increased by 7.9% and 14.4% for evergreen and deciduous forest until the 2090s, respectively. Evergreen forests were mainly simulated along the Himalayas, Western Ghats and eastern parts of the study region under current conditions (2000s, (Fig.3a), but expanded into

the south of peninsular India in future periods (2050s and 2090s) under RCP4.5+$eCO_2$. Deciduous forest cover also increased in future periods in central India and along the Himalayas (Figs.3 and S6).

The extent of $C_4$ savanna showed a significant decrease under scenarios with $eCO_2$, although in RCP4.5+$eCO_2$, it showed a increase by 12.1% between the 2010s and the 2050s (Table. 1, Fig.3). Simulated $C_4$ savanna area decreased by 14.1% relative to the 2000s until the 2090s under RCP4.5+$eCO_2$. Under RCP8.5+$eCO_2$ the model projected a decrease in $C_4$ savanna

area of 21.6% and 32.2% until the 2050s and the 2090s, respectively. The area covered by $C_4$ savanna increased under both RCPs with $fCO_2$ (Table.1). $C_4$ savannas were mainly located in the northern plain and peninsular India in the baseline period. However, these areas were replaced by deciduous forests in the northern plain and central India, and by evergreen forests in peninsular India and in the southeast of the region by the 2090s under $eCO_2$ scenarios (Figs.3a and S6a). The model simulated a decrease of area covered by woodland, shrubland, grasslands and $C_3$ savanna by the 2090s under all scenarios (Table. 1, Fig.

3). Simulations showed an increase in barren areas in the western part of the region under all scenarios (Figs. 3 and S6, Table. 1).

### 3.3   Projected changes in biomass at biome level

The aDGVM2 predicted an increase in mean biomass for evergreen and deciduous forest in the $eCO_2$ scenarios for both RCPs (Table. 2). Under RCP4.5+$eCO_2$, mean above ground biomass in evergreen and deciduous forest increased by 8.1% and 14.4%

by the 2050s and 3.8% and 15.7% by the 2090s, relative to the baseline period. The increase is even higher under RCP8.5+$eCO_2$ (Table. 2). The mean biomass of woodland decreased under both RCPs except for the 2050s with $eCO_2$ scenarios. The mean biomass of grassland increased under RCP4.5, but decreased for $C_4$ grassland under RCP8.5 for both $fCO_2$ and $eCO_2$ scenarios. Shrublands in the western part of the study region showed an increase in mean biomass under $eCO_2$ scenarios except for the 2050s under both RCPs, and a decrease under $fCO_2$ for both RCPs (Table. 2). Our results showed that under RCP4.5 and

RCP8.5 biomass decreased in the areas along the Himalayas, as well as in the central, north-eastern and western parts of the study region by the end of the century. Modelled biomass decrease is higher under RCP8.5 in these regions (Figs. 4 and S7). Biomass in the central and south-eastern part of the region is projected to increase under both RCPs with $eCO_2$ until the 2050s and 2090s, and to decrease in southern India and in parts of western South Asia (Figs. 4 and S7). We found increased biomass in Afghanistan, western Pakistan, Nepal and the southern part of Myanmar, and decreased biomass in the western arid part of

the study region under both RCPs for both $eCO_2$ and $fCO_2$ (Fig.5), though the magnitude of change is different (Figs.4 and S7). There were few areas in the western part of the study region where the model predicted increased biomass only under

$fCO_2$ for both RCPs (Figs.5). In large parts of the study region, biomass increased under $eCO_2$ for both RCPs but decreased under $fCO_2$, that is, $CO_2$ fertilization compensates climate change induced biomass die-backs in these regions (Figs.5).

### 3.4 Projected changes in evapotranspiration at biome level

The response of simulated $E_{biome}$ varies in different biomes under both RCP4.5 and RCP8.5 (Table. 3). Under the RCP4.5+$fCO_2$ scenario the model predicted a decrease in ET in all biomes except for deciduous forest and shrubland where it increased by 1% and 2.1% until the 2050s, and by 0.3% and 11.9% by the 2090s, respectively. Simulated $E_{biome}$ under RCP8.5+$fCO_2$ for deciduous forest and shrubland increased by 4.2% and 5.2% until the 2050s, and by 5.2% and 16.4% until the 2090s, respectively. The model also predicted increased $E_{biome}$ for $C_4$ grassland, evergreen forest and $C_4$ savanna until the 2090s
under RCP8.5+$fCO_2$ (Table.3). Comparisons of the RCP4.5+$fCO_2$ and RCP8.5+$fCO_2$ scenarios indicated that the former had a higher $E_{biome}$ than the latter scenario across all biomes because precipitation decrease is higher in the RCP8.5 scenario than in the RCP4.5 scenario. Under both RCPs with $eCO_2$, the model predicted a decrease in $E_{biome}$ across all biomes, except for a marginal increase in shrubland under RCP4.5 and deciduous forest under RCP8.5 until the 2050s and the 2090s (Table. 3). In general, scenarios with $eCO_2$ showed lower biome-specific evapotranspiration across ($E_{biome}$) most of the biomes compared to
simulations with $fCO_2$.

### 3.5 Response of mean ET and mean above ground biomass to climate change

The model predicted a larger increase in absolute annual mean ET (mm/year) under $eCO_2$ than $fCO_2$ for both RCP scenarios due to the corresponding increase in biomass (Figs. 4 and S7). We compared the spatially averaged annual values over entire South Asia of simulated absolute ET with MAP over the period from 1951 to 2099 and found a statically significant relation
(p-value <0.005). We found that absolute ET was positively correlated with MAP under all four scenarios (Figs.6a and S8a), but weakly correlated with MAT (Figs. 6b and S8b). For a given MAP, the spatially averaged annual value of above ground biomass (AGBM) was lower in scenarios with $fCO_2$ than scenarios with $eCO_2$ under both RCPs (Figs. 6c and S8c). The spatially averaged annual value of AGBM decreased beyond a MAT of 23.5°C for both RCPs with $fCO_2$, whereas it increased beyond 23.5°C under both RCP scenarios with $eCO_2$ (Figs.6d and S8d).

### 3.6 Impact of climate change on climatic niches of biomes

The climate niches of simulated biomes broadly overlapped with the biome niches in the Whittaker scheme (Figs. 7 and S9, Ricklefs, 2008; based on Whittaker, 1975). Under RCP4.5+$eCO_2$ and RCP8.5+$eCO_2$, the aDGVM2 simulated shifts of climatic niches of biomes. Evergreen and deciduous forest biomes were predicted to invade the niche space of savannas under $eCO_2$ scenarios (Figs.7 and S9). Savannas in turn were predicted to expand their climatic niche to MAT > 30°C by 2099, a climatic
space that was essentially not occupied by current biomes. Under RCP8.5+$eCO_2$ in the 2090s, forests completely occupied the climate space by 2090s which currently occupied by savanna (Fig. S9).

In both scenarios with fCO$_2$, savanna occupied the climate space delineated by MAT >25°C and MAP between 500mm and 1500mm and did not experience major replacement by forest. The model predicted that savanna expansion in climate space was higher under RCP8.5+fCO$_2$ than under RCP4.5+fCO$_2$ (Figs.7 and S9). Other biomes also experienced shifts in their climate space (Fig. 7). However, the results showed that for both current and future period grasslands and shrublands occupied the region with low MAP (<500mm), and woodland occupied low MAP (<800mm) regions, which corresponds to the western arid and semi-arid region of the study region under scenario with eCO$_2$ (Fig. 7).

## 4    Discussion

### 4.1    Impact of climate change and elevated CO$_2$ on biomes and biomass

Our simulations for RCP4.5+eCO$_2$ and RCP8.5+eCO$_2$ showed a strong positive response of vegetation growth, i.e., increases of biomass, canopy cover and canopy height. Mean biomass in most biomes was projected to increase, but the magnitude of increase differed considerably between different scenarios (Table.2). Projected change in canopy cover resulted in biome transitions. Under future conditions, the spatial distribution, extent and biomass of evergreen forests mostly remained at the current state, and evergreen forests were more resistant to climate change than deciduous forests. Expansion of deciduous forest into open biomes due to increasing woody cover resulted in significant loss of savanna area in the Deccan region under both RCPs with eCO$_2$ by the end of the century. Transition from deciduous forests to evergreen forest was simulated for the mountain regions of South Asia (Scheiter et al., 2020) i.e., the Himalayas and the Western Ghats, where precipitation was predicted to increase. The trade-offs between specific leaf area A$_{SLA}$ –leaf longevity (LL) results in emergence of evergreen behavior in wet regions South Asia. In the wet tropics, higher LL allows achieving a constant positive carbon balance from photosynthesis throughout the year and increases the residence time of nutrients and carbon in the plant and therefore enhances the photosynthetic gain per unit carbon and nutrient investment in leaves Kikuzawa and Lechowicz (2011).The deciduous behavior is advantageous in dry regions, as in Deccan region, because trees that do not invest much carbon into their leaves per unit dry mass (higher A$_{SLA}$ and lower LL) lose less investment when shedding them during the dry season. Phenology change as a result of climate change have already been observed (Buitenwerf et al., 2015; Cleland et al., 2007). In Scheiter et al. (2020), we showed that climate change supports transitions to tall evergreen vegetation in tropical Asia and found increases in the abundance evergreen plants and decreases in the abundance deciduous plants in mainland Southeast Asia, central India and Pakistan. This relative advantage of evergreen plants over deciduous plants under elevated CO$_2$ in aDGVM2 can be explained fact that increased intrinsic water use efficiency under eCO$_2$ in evergreen plants are higher than in deciduous plants as demonstrated by Soh et al. (2019). Previous modeling studies also support aDGVM2 result showing transitions from deciduous to evergreen vegetation . With the BIOME4 model, Ravindranath et al. (2006) simulated the response of forest to SRES A2 and B2 scenarios and reported similar changes toward evergreen phenology. A study by Chaturvedi et al. (2011) using the IBIS model also predicted transitions toward evergreen forest.

Woody encroachment in many ecosystems is attributed to rising $CO_2$ and this is supported by studies based on both field observations (e.g., FACE experiments) and satellite data (Brienen et al., 2015; Fischlin et al., 2007; Piao et al., 2006; Schimel et al., 2015; Archer et al., 2017; Stevens et al., 2017). The aDGVM2 also supports these findings i.e., increasing canopy cover and woody biomass under the $eCO_2$ condition and agrees with the reported greening trend in South Asia during the last three decades (Wang et al., 2017). Elevated $CO_2$ affects plants by increasing their photosynthetic rate, growth rate and water use efficiency, leading to an increase in biomass (Leakey et al., 2009; Norby and Zak, 2011). Increased photosynthetic rates under elevated $CO_2$ are due to an increase in the rate of rubisco carboxylation for $C_3$ plants, with a concurrent decrease of photorespiratory losses of carbon (Long et al., 2004). Due to the improved carboxylation efficiency, $C_3$ plants can respond by reducing stomatal conductance, thereby reducing transpirational losses, improving leaf water status, water use efficiency, and favoring leaf area growth (Long et al., 2004; Norby and Zak, 2011). Evidence from both observation and modelling of forest dynamic suggests that combined effects of $eCO_2$ and increased water use efficiency include increases in forest growth, canopy greening, wide spread increases in woody plant biomass and potential forest carbon sink. However, it is still unclear how the $CO_2$ responses scale to the ecosystem level (Hickler et al., 2015), and how nutrient limitation from the soil may influence ecosystem responses to $eCO_2$. Körner (2015) argued that carbon from atmosphere can only be converted into biomass if other factors such as nutrients, temperature and water are not limiting. In addition, benefit of $eCO_2$ can be down-regulated by broad scale forest die-off due to frequent drought and warmer temperature (Choat et al., 2018; Mcdowell et al., 2016), tree mortality due negative tree physiological responses and negative carbon balance and accelerated pest attacks. Rising background mortality rates combined with projected increases in intensity, frequency and duration of drought (Huang et al., 2016) increases the uncertainty regarding positive effect of $eCO_2$.

In the long run, whether ecosystems act as carbon source or sink can be estimated using models that consider all factors that are relevant in the carbon cycle and its associated factors (Fatichi et al., 2014; Körner, 2015). However, (Terrer et al., 2019), showed that the global-scale response to $eCO_2$ from experiments is similar to past changes in greenness (Piao et al., 2019) and biomass (Sitch et al., 2015) in response to $eCO_2$. This suggests that $CO_2$ will likely continue to stimulate plant biomass in the future despite the constraining effect of soil nutrients, however also argued that the empirical relationships with soil nutrients can be powerful for explaining large-scale patterns of $eCO_2$ responses, despite ecosystem-level uncertainties. According to our simulations we can conclude that natural vegetation of South Asia likely will remain a carbon sink in the future (Fig.5).

## 4.2 Impact of climate change and fixed $CO_2$ on biomes and biomass

Under both $fCO_2$ scenarios, the spatial distribution of savanna areas remained in its contemporary state. Central India and the Deccan Plateau showed a transition of deciduous forest to savanna, because forest canopy opened up due to tree mortality caused by increasing temperature and reduced MAP. This indicates that plants experience temperature and drought stress under fixed $CO_2$. These stresses were compensated by $CO_2$ fertilization in $eCO_2$ scenarios where the aDGVM2 simulated increased biomass and woody encroachment in areas affected by climate-induced die-back in $fCO_2$ simulations. This aDGVM2 behavior agrees with results presented by Lapola et al. (2009) who modelled biome shifts from forest to savanna in absence of $CO_2$ fer-

tilization for the Amazon region. Changes in precipitation regimes are likely to have a strong influence particularly in arid and semi-arid regions, such as grasslands (Verstraete et al., 2009). The complex interactions of inter-annual precipitation variability and precipitation seasonality can result in rapid ecosystem transitions (e.g., between alternative stable states with high and low vegetation biomass; Holmgren and Scheffer, 2001). The decrease in simulated AGBM after MAT increases beyond 23.5°C under $fCO_2$ scenarios can be explained by the longer exposure of vegetation to temperatures beyond the optimum temperature range of $C_3$ photosynthesis during the main growing season. This effect was further enhanced by decreasing MAP and the absence of $CO_2$ fertilization. This implies that the increase in MAT above 23.5°C together with weak $CO_2$ fertilization would have negative consequences for carbon sequestration. The sensitivity of biomass to temperature and $CO_2$ change has been investigated in many studies (Norby and Luo, 2004; Song et al., 2019; Sperry et al., 2019). A meta-analysis by Lin et al. (2010) showed that warming significantly increased biomass by 12.3% (with a 95% confidence interval of 8.4–16.3%) across all the terrestrial plants included. This observation is consistent with our model results. Biomass showed a positive relation with MAT which did not change with mean annual precipitation or experimental duration or CO2 enrichment (Lin et al., 2010). These findings are also supported by previous studies by Rustad et al. (2001), Walker et al. (2006), and Dormann and Woodin (2002) which have revealed that warming generally increases terrestrial plant biomass, indicating enhanced terrestrial carbon uptake via plant growth. Previous modeling studies using Biome-BGC (Running and Hunt Jr, 1993), Century (Parton et al., 1993), and TEM Tian et al. (1999) have shown an increase in productivity when both climate change and $CO_2$ effects were considered. However, the increase was smaller when only climate change effects were considered and both Biome-BGC and TEM suggest that without $CO_2$ fertilization, average productivity would decline relative to current annual average as shown by our result (Fig. 6d). This suggests complexity and challenges in seeking general patterns of terrestrial plant growth in a future warmer climate condition. It also implies that we need a better understanding of impacts of heat stress on vegetation and how it interacts with drought and $CO_2$ fertilization. It is also unclear to what degree thermal acclimation may counteract some of the negative effects on plant growth caused by higher temperatures (Lombardozzi et al., 2015).

## 4.3 Impact of climate change and $CO_2$ change on climatic niches of biomes

Elevated $CO_2$ has a major impact on the climatic niche space of biomes. Our simulations showed forest invasion into the niche space currently occupied by savanna by the end of the century. The expansion of forests to drier areas corresponds to a widening of their climate niche space under $eCO_2$. This expansion is mainly driven by $eCO_2$ is corroborated by the fact that in absence of $CO_2$ fertilization the climatic niche of biomes is stable, i.e., biomes occupy the same niche space under current and future conditions. These findings imply that the bioclimatic boundaries used to define biome niche space are not static, but are specific for given $CO_2$ levels. Therefore, the thresholds of the Whittaker's biomes need to be redefined for a high-$CO_2$ world such that they encompass the altered climatic envelopes of biomes under elevated $CO_2$ in future (Fig. 7). The shift in niche space can be attributed to the shift in plant communities caused by the combined effect of climate change and elevated $CO_2$, which increases plant water use efficiency allowing them to grow under drier conditions. These community shifts can also lead to change the characteristics of biomes by altering community structure and ecosystem functions (Chapin et al., 1997).

## 4.4  Effect of $CO_2$ on ET and its interaction with climate change

Climate change has direct effects on hydrological processes (Liu et al., 2008). ET and water deficit influence plant productivity and distribution (Stephenson, 1998). Higher biomass coincided with increased absolute amounts of ET for $eCO_2$ scenarios in some parts of the study region under both RCPs by the end 21st century (Figs. 5 and S7). This change can be attributed to higher leaf biomass accumulated in plants ebaled by increased photosynthetic efficiency under $eCO_2$. The higher amount of leaf biomass offsets the water-saving effect caused by reduced stomatal conductance due to improved water use efficiency under $eCO_2$ scenarios and resulted in reduced ET per unit leaf biomass (Warren et al., 2011). Our results showed that the strength of the $CO_2$ fertilization effect is relevant when attempting to determine $E_{biome}$ at biome level during the 21st century. Biome-specific ET decrease was less pronounced under RCP4.5 due to a lower concentration of atmospheric $CO_2$ compared to RCP8.5. Our simulated decrease in ET in response to climate change and increasing $CO_2$ concentration agrees with Kergoat et al. (2002) who have reported decreased ET under elevated $CO_2$ concentration in a chamber experiment. However, reduced ET under $eCO_2$ can reduce regional-scale atmospheric humidity and thereby enhance the vapor pressure deficit (VPD), between leaves and atmosphere, a driving force for water loss, which may partially counteract $CO_2$-induced reduction of ET due to decreased stomatal conductance. Due to stomatal closure, photosynthetic rates under soil water stress conditions decline in aDGVM2 when atmospheric VPD increases. Projected increase in air temperature also increases the saturated water vapor pressure. As a result VPD will increase, given that increase in actual vapor pressure is limited by soil water availability whereas the increase in saturated vapor pressure is not (Yuan et al., 2019) and potential evapotranspiration will increase with temperature (Warren et al., 2011). As future climate projections vary spatially and temporally, there was high model uncertainty on how ET will respond to changes in precipitation and temperature.

## 4.5  Implication of the projected change for conservation

Changes in biome types imply changes in biodiversity, ecosystem function and productivity. Each biome is characterized by a range of distinctive ecological processes and functions. For instance distribution of forest ecosystem in the mountains is largely regulated by the altitude and climatic factor (Saikia et al., 2017). They have high species richness and needed to be protected from the ever-increasing anthropogenic pressure and climate change. Open biomes such as grassland and savanna support high biodiversity (Parr et al., 2014). Pronounced increases in tree density in grasslands and savannas will alter vegetation structure and reduce grassland biodiversity. Such changes will negatively affect savanna-specific ecosystem services such as grazing potential, tourism and wildlife habitat availability (Parr et al., 2012). The threat posed to the biodiversity of Asian savannas by climate change is aggravated by inadequate management policies that misinterpret them as degraded forest (Ratnam et al., 2016). In this context, management policies aim to afforest open biomes, although paleo-ecological evidence indicates that these open biomes are not degraded forest but ancient ecosystems (Kumar et al., 2020; Ratnam et al., 2016). Moreover, increased woody cover can negatively affect water resources in the semi-arid regions of the study area. Acharya et al. (2018) have shown that increased woody cover hinders the downward movement of water causing increased water inception which have negative effects on ground water recharge. It is therefore necessary to control the abundance of woody plants in semi-arid

regions to control stream flow and enhance groundwater recharge (Bednarz et al., 2001).

In South Asia, biodiversity hotspots have a very unique topography, where climate varies strongly over short distances. As global biodiversity hotspots, mountain forest ecosystems in the Western Ghats, the Himalayas and northeastern part of the study area (Indo-Myanmar) are particularly vulnerable to climate change (Myers et al., 2000) and need targeted management action to mitigate adverse effects. Conservation of these hotspots requires consideration of many different attributes of plant

communities, ecosystems, landscapes, and plant diversity, how they will change, and how their ecosystem services are valued.

Conservation methods and policies that can accommodate minimal losses of ecosystem services and provide robust strategies to mitigate climate change impacts should be developed and implemented. In this context, DGVMs facilitate the exploration of vegetation-climate interactions by providing detailed results for different management and climate scenarios. Such an ex-

520 ploration of different possible scenarios is necessary to develop optimized mitigation and conservation strategies for protected areas in biodiversity hotspots. The value of DGVM modelling results lies in their potential to provide insights into multiple future trajectories. Based on the most likely trajectories, the results can be used to tailor best-practice strategies for decision makers that need to manage conservation areas or protected areas Boulangeat et al. (2012).

### 4.6 Limitations of this modelling study

Our simulation results are constrained by the model formulation and the assumptions underlying aDGVM2. Disagreement between model results and data used for benchmarking can be attributed to the fact that the aDGVM2 simulates potential natural vegetation whereas remote sensing products integrate land use. This implies that enhancing the model to simulate observed land cover patterns would require additional information on anthropogenic impacts. Anthropogenic activities such as deforestation, habitat conversion and urbanization can modify the interactions between climate, plant communities and biomes

(Hansen et al., 2001).

In addition data-model disagreement can be explained by model uncertainties and processes currently not considered in aDGVM2. For instance, aDGVM2 uses carbon allocation parameters that are not easily measurable in the field, which limits the evaluation of simulated mechanisms. The model currently does not account for carbon that plants invest into nutrient acquisition (e.g. mycorrhiza) or into defences against predation and pathogens (Zemunik et al., 2015). There is insufficient

ecophysiological data from the study region, required for parameterization of trait ranges used to simulate rgional plant communities (Kumar and Scheiter, 2019). The complexity of the interactions between global change and biomes as well as biodiversity is difficult to model in absence of such data. While the model currently capture the more optimistic effects related to $CO_2$ fertilization and temperature, associated mortality reasons such as pests attack, heat damage to plant tissues, etc are insufficiently represented in the models. The low resolution of input data both soil and climate data also limits the model's

capability to capture high resolution regional heterogeneity in vegetation distribution. Further, the strength of $CO_2$ fertilization modelled in aDGVM2 may be overestimated because the effect of nutrient limitation on productivity is not included in this

version of aDGVM2 (Körner et al., 2005; Terrer et al., 2018). Despite these caveats, we are nonetheless confident to capture general patterns of future global change and its consequences for biomes and their boundaries in South Asia.

## 5   Conclusions

The model reproduced the contemporary distribution of biomes, biomass, evapotranspiraton and tree height. We investigated the impact of $eCO_2$ and climate change on South Asian biomes and found that climate change and $CO_2$ fertilization in combination are substantial drivers of biome change, and that elevated $CO_2$ concentrations altered the climatic envelope of biomes in addition to causing increases in biomass, tree height and canopy cover. Continued biomass increase indicates that South Asia's natural vegetation will likely remain a carbon sink in the 21st century. Our results also imply that woody encroachment poses

threat to open biomes and causes transition of savanna biomes to deciduous forest in the future. The savanna biome is important in the context of biodiversity conservation. We showed that bioclimatic niches of biomes are not static, but are specific for given $CO_2$ concentrations. We therefore argue that Whittaker plots used top illustrate niches of biomes need to be adjusted for future climate condition. We also found that simulated decrease in biomass-specific ET is more pronounced in scenarios with $eCO_2$ than in scenarios with $fCO_2$ which indicates that water use efficiency will likely increase due to $CO_2$ fertilization.

The biome transitions simulated under $eCO_2$ and changing climate indicate the need to adjust ecosystem management, mitigation strategies, and conservation policies for protected areas to allow targeted long-term management. To understand the significance of ecological responses to climate change, it is essential to improve and expand biological monitoring activities (Loreau et al., 2001). To achieve this, the most vulnerable biomes that we identified in this study could be proposed as high-priority targets for programs that monitor vegetation-climate interactions, productivity and biodiversity Proença et al. (2017).

*Code availability.*   The aDGVM2 code as well as scripts to conduct the model experiments and analyze the results are available upon request. Please contact any of the authors.

*Author contributions.*   DK and SS conceived the study, DK included changes to aDGVM2 for the study region, conducted model simulations, analyzed results, created figures and lead the writing. All authors contributed to the development of aDGVM2 and commented to the text.

*Competing interests.*   The authors declare that they have no conflict of interest.

*Disclaimer.*   Key words: aDGVM2, climate change, CO2-fertilization, biome shift, woody encroachment, evapotranspiration, biodiversity conservation, South Asia

*Financial support.* SS and DK thank the Deutsche Forschungsgemeinschaft (DFG) for funding (Emmy Noether grant SCHE 1719/2-1). MP acknowledges funding by the German Federal Ministry of Education and Research (BMBF, SPACES2 initiative, 'SALLnet' project, grant 01LL1802B).

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

**Table 1.** Biome cover (in %) for the 2000s, 2050s and 2090s, and percent (%) change in biome cover from the 2000s to the 2050s and the 2000s to the 2090s under RCP4.5 and RCP8.5 with fixed and elevated $CO_2$. $\Delta$ indicates percent change in biome cover between time periods.

| RCP Scenarios | Year | Barren | $C_4$ Grassland | $C_3$ Grassland | Woodland | Evg. forest | Dec. forest | Shrubland | $C_4$ Savanna | $C_3$ Savanna |
|---|---|---|---|---|---|---|---|---|---|---|
| RCP4.5 + $fCO_2$ | 2010s | 5.6 | 15.4 | 4.6 | 18.2 | 11.7 | 17.6 | 6.9 | 17.7 | 2.4 |
| | 2050s | 6.3 | 14.8 | 3.2 | 15.7 | 11.2 | 18.6 | 6.7 | 22.1 | 1.4 |
| | 2090s | 10.4 | 12.3 | 2.3 | 10.0 | 12.7 | 20.1 | 6.0 | 24.7 | 1.4 |
| | $\Delta$ 2050s-2010s | 13.0 | -3.7 | -32.2 | -13.6 | -4.0 | 5.6 | -3.0 | 25.4 | -39.1 |
| | $\Delta$ 2090s-2010s | 87.0 | -20.1 | -50.0 | -45.2 | 7.9 | 14.4 | -12.7 | 40.1 | -41.3 |
| RCP4.5+ $eCO_2$ | 2010s | 5.7 | 15.2 | 4.8 | 18.6 | 11.5 | 17.5 | 6.8 | 17.5 | 2.4 |
| | 2050s | 6.5 | 13.9 | 3.5 | 15.0 | 11.9 | 21.2 | 7.0 | 19.6 | 1.3 |
| | 2090s | 10.4 | 10.4 | 2.5 | 10.7 | 15.9 | 27.9 | 6.2 | 15.1 | 0.9 |
| | $\Delta$ 2050s-2010s | 13.5 | -8.2 | -26.9 | -19.7 | 3.1 | 21.2 | 3.8 | 12.1 | -44.7 |
| | $\Delta$ 2090s-2010s | 82.0 | -31.6 | -48.4 | -42.4 | 38.0 | 59.1 | -8.4 | -14.1 | -63.8 |
| RCP8.5 + $fCO_2$ | 2010s | 6.3 | 14.7 | 4.5 | 18.8 | 11.7 | 17.3 | 6.3 | 18.0 | 2.4 |
| | 2050s | 8.8 | 12.3 | 2.5 | 14.7 | 12.9 | 21.7 | 6.6 | 19.0 | 1.5 |
| | 2090s | 9.4 | 15.0 | 2.5 | 11.0 | 10.8 | 14.2 | 6.7 | 29.0 | 1.4 |
| | $\Delta$ 2050s-2010s | 39.0 | -16.5 | -43.7 | -21.9 | 10.1 | 25.0 | 4.1 | 5.4 | -39.1 |
| | $\Delta$ 2090s-2010s | 48.8 | 1.8 | -43.7 | -41.6 | -7.9 | -17.9 | 5.7 | 61.0 | -41.3 |
| RCP8.5 + $eCO_2$ | 2010s | 5.9 | 14.8 | 4.7 | 18.0 | 11.6 | 17.5 | 7.1 | 17.9 | 2.5 |
| | 2050s | 9.7 | 10.5 | 3.2 | 13.9 | 14.5 | 25.4 | 7.1 | 14.1 | 1.6 |
| | 2090s | 6.3 | 11.5 | 4.2 | 12.6 | 17.0 | 28.0 | 7.0 | 12.2 | 1.3 |
| | $\Delta$ 2050s-2010s | 64.9 | -29.5 | -32.6 | -22.9 | 24.8 | 45.4 | 0.7 | -21.6 | -35.4 |
| | $\Delta$ 2090s-2010s | 7.0 | -22.2 | -10.9 | -30.3 | 46.5 | 60.2 | -1.5 | -32.2 | -47.9 |

**Table 2.** Mean biomass (in t/ha) within biomes for the 2000s, 2050s and 2090s, and percent (%) change in biomass from the 2000s to the 2050s and the 2000s to the 2090s under RCP4.5 and RCP8.5 with fixed and elevated $CO_2$. $\Delta$ indicates percentual biomass changes between time periods.

| RCP Scenarios | Year | $C_4$ Grassland | $C_3$ Grassland | Woodland | Evg. forest | Dec. forest | Shrubland | $C_4$ Savanna | $C_3$ Savanna |
|---|---|---|---|---|---|---|---|---|---|
| RCP4.5 + fCO₂ | 2010s | 0.9 | 1.5 | 30.4 | 189.7 | 142.1 | 4.0 | 35.5 | 36.8 |
|  | 2050s | 0.9 | 1.8 | 29.2 | 191.0 | 144.0 | 3.6 | 38.0 | 44.7 |
|  | 2090s | 0.9 | 2.1 | 24.5 | 188.1 | 148.4 | 3.3 | 32.6 | 31.8 |
|  | $\Delta$ 2050s-2010s | -1.1 | 19.5 | -4.0 | 0.7 | 1.3 | -10.9 | 6.8 | 21.4 |
|  | $\Delta$ 2090s-2010s | 4.4 | 35.1 | -19.4 | -0.9 | 4.4 | -17.8 | -8.2 | -13.7 |
| RCP4.5 + eCO₂ | 2010s | 0.9 | 1.4 | 30.7 | 189.2 | 142.5 | 4.0 | 35.9 | 37.3 |
|  | 2050s | 1.0 | 1.5 | 34.7 | 204.6 | 162.9 | 4.3 | 48.1 | 53.2 |
|  | 2090s | 1.0 | 1.6 | 29.3 | 196.4 | 164.9 | 4.1 | 43.2 | 51.8 |
|  | $\Delta$ 2050s-2010s | 17.2 | 5.6 | 13.0 | 8.1 | 14.4 | 6.0 | 34.0 | 42.7 |
|  | $\Delta$ 2090s-2010s | 12.6 | 8.3 | -4.6 | 3.8 | 15.7 | 2.5 | 20.4 | 39.1 |
| RCP8.5 + fCO₂ | 2010s | 0.9 | 1.5 | 30.7 | 191.1 | 146.3 | 3.9 | 35.8 | 34.9 |
|  | 2050s | 0.7 | 1.6 | 23.5 | 182.1 | 134.7 | 3.3 | 31.2 | 28.0 |
|  | 2090s | 0.8 | 1.6 | 18.7 | 175.7 | 136.4 | 3.1 | 28.5 | 33.2 |
|  | $\Delta$ 2050s-2010s | -19.1 | 4.7 | -23.4 | -4.7 | -7.9 | -15.3 | -12.8 | -19.7 |
|  | $\Delta$ 2090s-2010s | -14.6 | 4.7 | -39.0 | -8.0 | -6.8 | -20.0 | -20.5 | -4.9 |
| RCP8.5 + eCO₂ | 2010s | 0.9 | 1.3 | 31.2 | 188.3 | 146.1 | 4.1 | 36.5 | 32.0 |
|  | 2050s | 1.0 | 1.4 | 32.1 | 206.3 | 162.7 | 4.0 | 45.1 | 47.2 |
|  | 2090s | 0.7 | 1.1 | 30.8 | 206.0 | 183.4 | 4.7 | 49.8 | 50.7 |
|  | $\Delta$ 2050s-2010s | 9.9 | 8.7 | 2.8 | 9.6 | 11.3 | -1.5 | 23.6 | 47.4 |
|  | $\Delta$ 2090s-2010s | -22.0 | -12.7 | -1.6 | 9.4 | 25.6 | 15.5 | 36.6 | 58.2 |

**Table 3.** Biome-level ET normalized to biomass ($E_{biomes}$, mm/kg/year) for the 2000s, 2050s and 2090s, and percent (%) change in $E_{biomes}$ from the 2000s to the 2050s and the 2000s to the 2090s under RCP4.5 and RCP8.5 with fixed and elevated $CO_2$. $\Delta$ indicates percentual ET changes between time periods.

| RCP Scenarios | Year | $C_4$ Grassland | $C_3$ Grassland | Woodland | Evg. forest | Dec. forest | Shrubland | $C_4$ Savanna | $C_3$ Savanna |
|---|---|---|---|---|---|---|---|---|---|
| RCP4.5 + $fCO_2$ | 2010s | 186.7 | 95.5 | 257 | 159.7 | 288.5 | 183.3 | 252.5 | 194.2 |
| | 2050s | 170.9 | 80.5 | 217 | 157.4 | 291.3 | 187.2 | 244.6 | 151.9 |
| | 2090s | 185 | 72.3 | 209.6 | 140.7 | 289.3 | 205.2 | 247.1 | 179.1 |
| | $\Delta$ 2050s-2010s | -8.5 | -15.7 | -15.6 | -1.4 | 1 | 2.1 | -3.1 | -21.8 |
| | $\Delta$ 2090s-2010s | -0.9 | -24.3 | -18.5 | -11.9 | 0.3 | 11.9 | -2.1 | -7.8 |
| RCP4.5 + $eCO_2$ | 2010s | 185.4 | 93.4 | 259.7 | 159.7 | 288.1 | 190.9 | 251.6 | 188.4 |
| | 2050s | 161.2 | 79.7 | 217 | 147.8 | 283 | 183.2 | 238.2 | 153.4 |
| | 2090s | 164.1 | 73.4 | 210.2 | 138.7 | 280.4 | 197.2 | 236.6 | 157.1 |
| | $\Delta$ 2050s-2010s | -13.1 | -14.6 | -16.5 | -7.4 | -1.8 | -4.1 | -5.3 | -18.6 |
| | $\Delta$ 2090s-2010s | -11.5 | -21.4 | -19.1 | -13.2 | -2.7 | 3.3 | -6 | -16.6 |
| RCP8.5 + $fCO_2$ | 2010s | 172.8 | 87.4 | 257.5 | 160.9 | 286.5 | 185.5 | 244.7 | 188.1 |
| | 2050s | 153.7 | 72.7 | 243.2 | 158.3 | 298.5 | 195.1 | 241 | 162.7 |
| | 2090s | 195.6 | 67.6 | 231.1 | 162.7 | 301.3 | 216 | 267.5 | 150.2 |
| | $\Delta$ 2050s-2010s | -11.1 | -16.8 | -5.5 | -1.6 | 4.2 | 5.2 | -1.5 | -13.5 |
| | $\Delta$ 2090s-2010s | 13.2 | -22.6 | -10.2 | 1.1 | 5.2 | 16.4 | 9.3 | -20.1 |
| RCP8.5 + $eCO_2$ | 2010s | 177.5 | 91.1 | 256.4 | 162.7 | 284.5 | 192.5 | 243.7 | 191.7 |
| | 2050s | 143.9 | 76.9 | 235.6 | 149.4 | 285.4 | 184.6 | 228.8 | 153.1 |
| | 2090s | 141.4 | 59.2 | 218.3 | 143.9 | 284.9 | 186 | 242.3 | 143.2 |
| | $\Delta$ 2050s-2010s | -18.9 | -15.6 | -8.1 | -8.1 | 0.3 | -4.1 | -6.1 | -20.1 |
| | $\Delta$ 2090s-2010s | -20.3 | -35.1 | -14.9 | -11.6 | 0.1 | -3.4 | -0.6 | -25.3 |

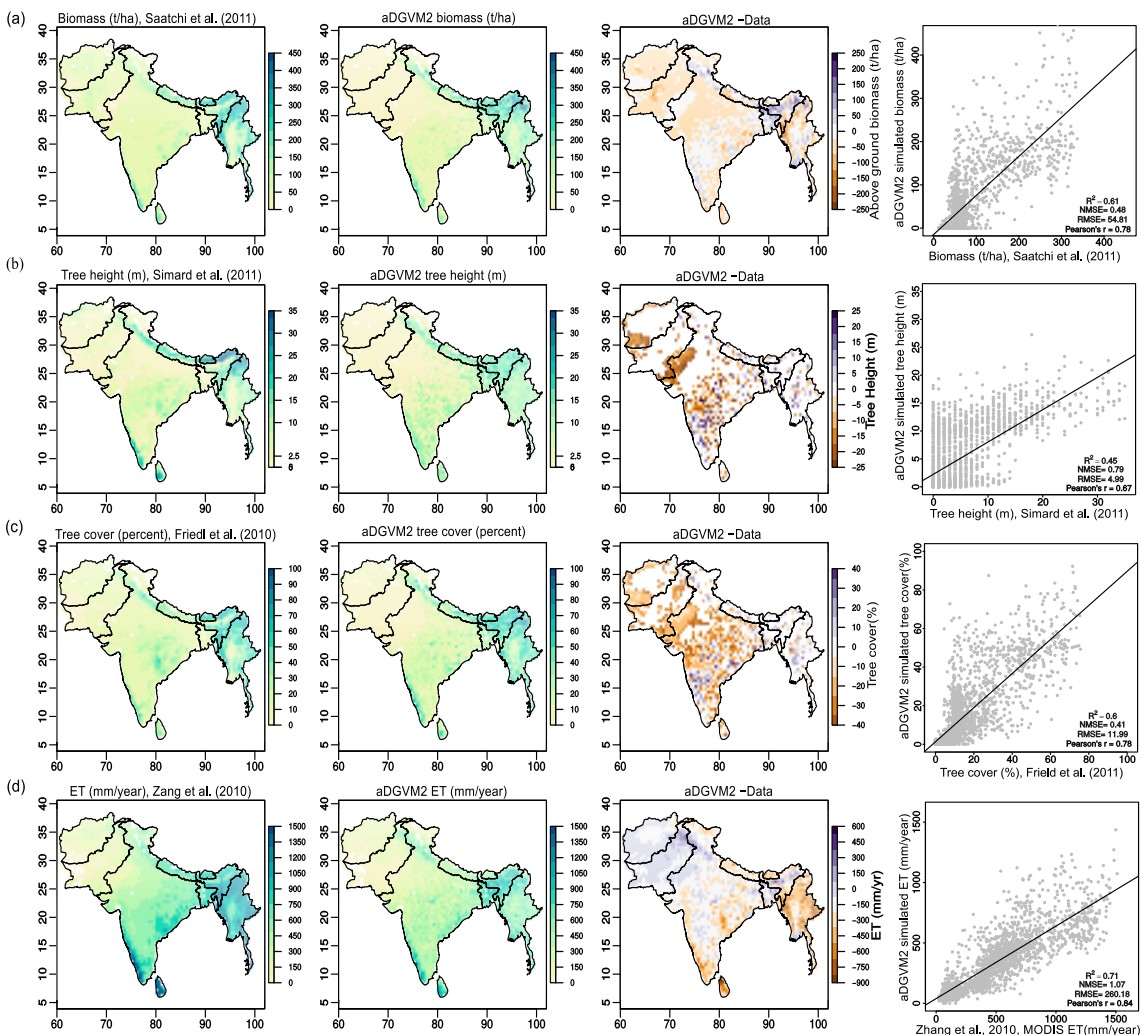

**Figure 1.** : Comparison between aDGVM2 results and data for (a) simulated biomass and Saatchi et al. (2011) biomass, (b) simulated tree height and Simard et al. (2011), (c) simulated tree cover and Friedl et al. (2011) tree cover and (d) simulated evapotranspiration and Zang et al. (2010) evapotranspiration. In the figure the first column shows the remote sensing products, the second column shows aDGVM2 results, and the third column shows the difference between simulation and data and the fourth column shows the scatter plot between simulated state variables against benchmarking data. NMSE and RMSE are normalized mean square error and root mean square error, respectively. In the fourth column, each point represents one grid cell in the study region. For results with masked land use cover see supplementary Figure S4.

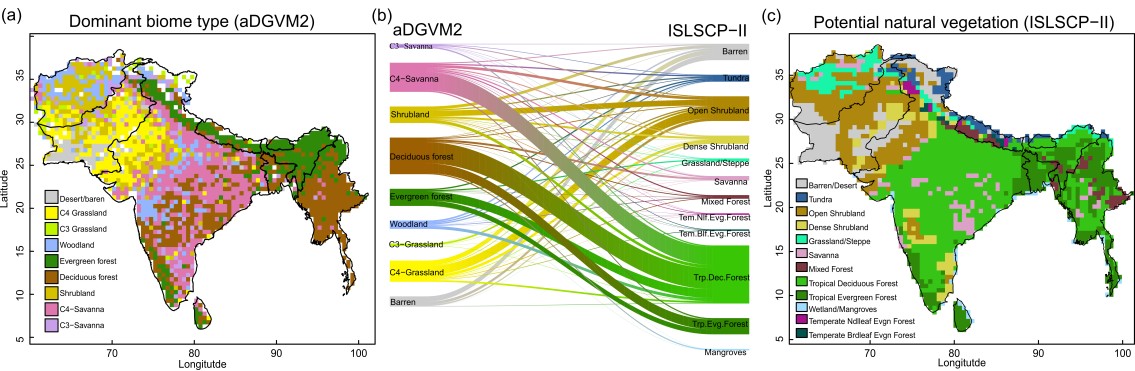

**Figure 2.** Comparison between simulated and observed biome patterns. (a) Simulated dominant biome type, (b) Sankey diagram showing overlap between simulated biomes and potential natural vegetation cover (ISLSCP-II, Ramankutty et al., 2010) and (c) potential natural vegetation. The Sankey graph shows how aDGVM2 biomes and PNV classes overlap.

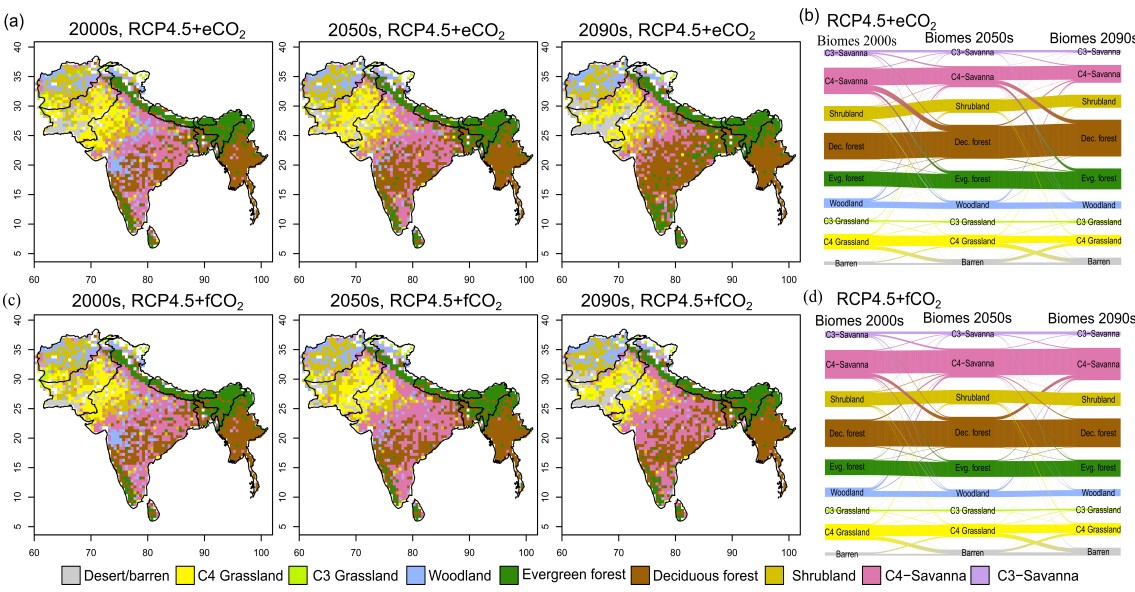

**Figure 3.** Simulated biome distribution for the 2000s, 2050s and 2090s under (a) RCP4.5+eCO$_2$ and (c) RCP4.5+fCO$_2$.The Sankey diagrams showing the fractional cover of biomes and transitions between biomes from the 2000s to the 2050s and the 2050s to the 2090s under (b) RCP4.5+eCO$_2$ and (d) RCP4.5+fCO$_2$. See Figure S6 for simulated biome distribution under RCP8.5.

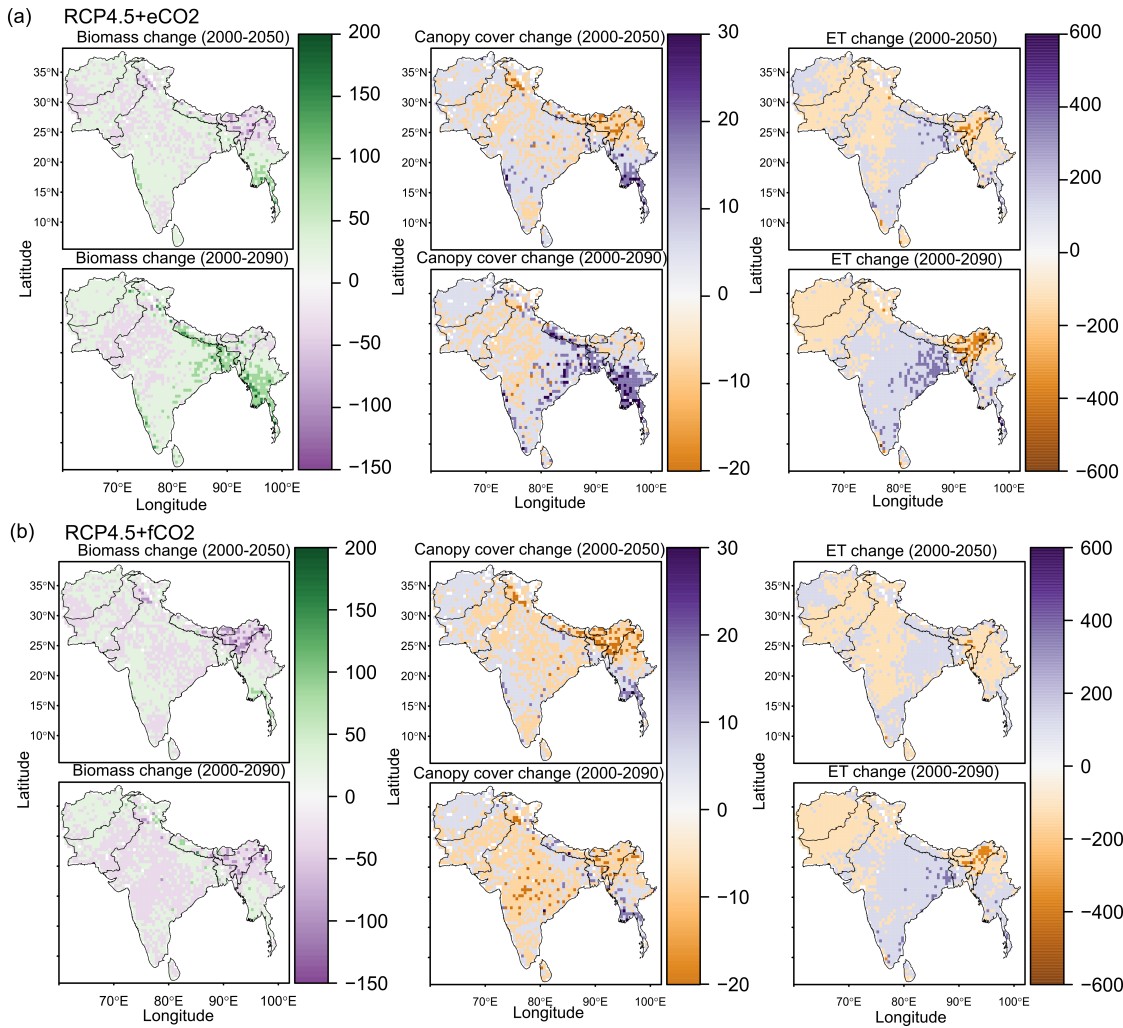

**Figure 4.** Projected change in biomass (t/ha), canopy area (%) and evapotranspiration (ET, mm/year) between the 2000s and 2050s, and between the 2000s and the 2090s under (a) RCP4.5+eCO$_2$ and (b) RCP4.5+fCO$_2$. See Figure S7 for projected change of these variables under RCP8.5.

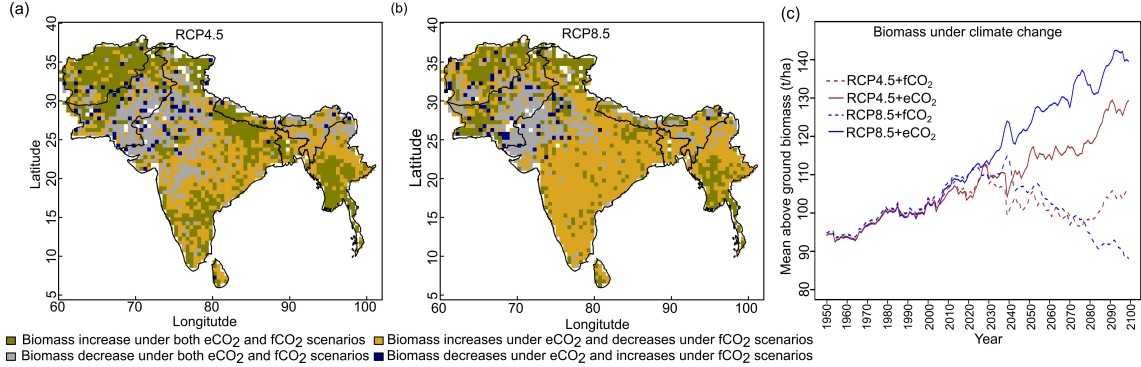

**Figure 5.** Maps showing areas where $CO_2$-fertilization compensates biomass dieback caused by climate change between the 2000s and the 2090s under (a) RCP4.5 and (b) RCP8.5. and (c) aboveground biomass between 1950 and 2099 for South Asia in different scenarios.

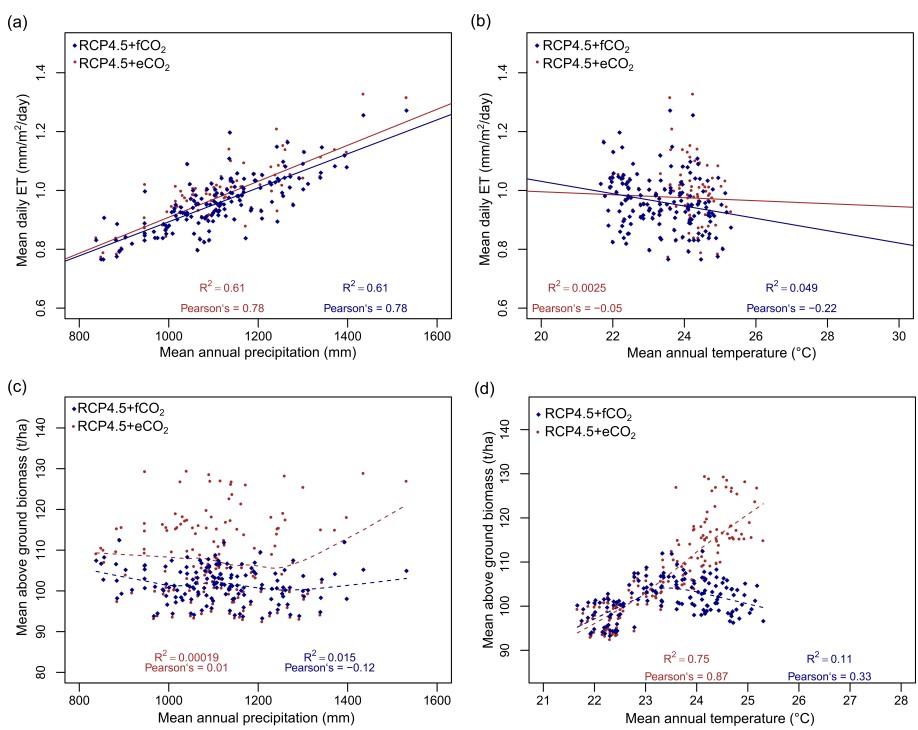

**Figure 6.** Relationship between (a) evapotranspiration (ET) and mean annual precipitation (MAP), (b) ET and mean annual temperature (MAT), (c) mean above ground biomass and MAP and (d) mean above ground biomass and MAT under RCP4.5. The lines (both solid and dotted) in all figures represent the best-fit regression line. Data points represent spatially averaged ET (a, b) and biomass (c, d) over entire South Asia for each year from 1950 to 2099. See Figure S8 for results under RCP8.5.

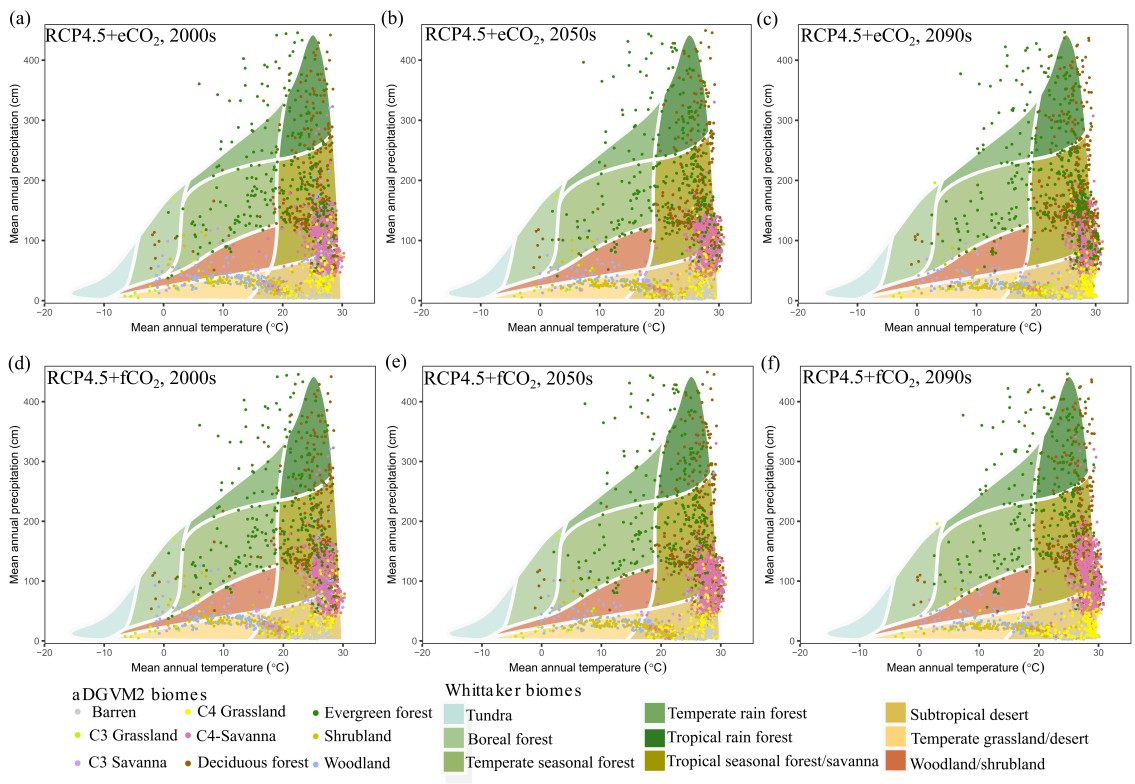

**Figure 7.** Simulated climate niches of biomes for the (a) 2000s, (b) 2050s and (c) 2090s under RCP4.5+eCO$_2$ and (d) 2000s, (e) 2050s and (f) 2090s under RCP4.5+fCO$_2$. The simulated biomes are overlaid on the climate envelopes of Whittaker's biomes and are plotted following Ricklefs (2008) and Whittaker (1975). See Figure S9 for projected change in climatic niches of biomes under RCP8.5.