# Peer review of "Climate change and elevated CO2 favor forest over savanna under different future scenarios in South Asia"

_Biogeosciences, 2020_

## Referee Comment (RC1) · Anonymous Referee #1 · 29 Jun 2020

Dear Editor,

Review of the manuscript "Climate change and elevated CO2 favour forest over savanna under different future scenarios in South Asia.

The manuscripts read well and has the most relevance topic covered for the day given we are experiencing unprecedented climate change impacts and resultant pandemic. The aim to look into this south Asian scenario on forest biomass and changing forest cover over the region given the major of the population are dependent on natural resources for their livelihoods. No doubt these models developed point at the future direction and provide policy makers sufficient time to plan and execute appropriate

policies aiming to conserve limited natural resources and protect the livelihoods of millions.

The manuscript postulates that elevated CO2 levels improve the C3 function to enhance biomass productivity, which is quite simple to understand and model based on this assumption. However, at molecular level, the RUBP carboxylase has a different function based on the CO2 concentration, temperature and sunlight. Photosynthetic enzyme RUBISCO has dual function that fixes carbon when there is higher concentration of CO2 while at arid situations may act as RUBP oxygenase i.e., reduce carbon instead of fixing it. Thus, I request authors to go through the following articles which provide molecular basis of carbon fixation and how this might be helpful to model the scenarios under climate change.

1. James R Ehleringer and Thure E Cerling 2002, C3 and C4 Photosynthesis 2002 Volume 2, The Earth system: biological and ecological dimensions of global environmental change, pp 186–190 Edited by Professor Harold A Mooney and Dr Josep G Canadell in Encyclopedia of Global Environmental Change (ISBN 0-471-97796-9) Editor-in-Chief Ted Munn, John Wiley & Sons, Ltd, Chichester https://d1wqtxts1xzle7.cloudfront.net/38426075/C3_ad_C4_photosynthesis.pdf?1439136433=&response-content-disposition=inline%3B+filename%3DC3_and_C4_Photosynthesis.pdf&Expires=1593404295&Signature=aSHD6C 2uvohaB9ocTzzsdtH2Gg0pkJ3elFWyEmFKxx8GPH∼hbRUY8GEXa-3HwNWKtzR33dLW∼hGJlb-WPoisOwxVaU227JcrqW8SRNTEQx3-wbiRVSw4hMBzhwY7heqUnffNwPy7avhdQnZQ1DFRR5DxBGvSP76vxWFmkxNTpDiRC4TP8KCRJM2YZZTH04a5LqXC Pair-Id=APKAJLOHF5GGSLRBV4ZA 2. Ian E. Woodrow and Joseph A. Berry 1988 ENZYMATIC REGULATION OF PHOTOSYNTHETIC CO2 FIXATION IN C3 PLANTS Ann. Rev. Plant Physiol. Plant Mol. Bioi. 1988. 39:533-94 https://www.annualreviews.org/doi/pdf/10.1146/annurev.pp.39.060188.002533

Furthermore, the manuscript under discussion line 349 (page 12) predicts expansion of woody forests compared to Savannas in mountainous regions. Similarly predicts that in Deccan plateau may change to evergreen forests from deciduous forests. This appears

a very sweeping statement, given there is variation in rainfall associated with increasing temperatures Deccan region of the south Asia may experience floods and frequent droughts. It is well established that soils in Deccan region are low in nitrogen and other micro-nutrients, which support higher biomass production. I am not sure whether such physical attributes were considered while modelling the forest cover change. In fact, the regions of the deccan plateau currently experience severe droughts which may cast uncertainty of the results obtained by the Authors. Under these situations the conclusions drawn seems to be quite simplistic without applying appropriate knowledge in the forestry sector. The authors at page 13 line 376 explain aptly that decreased biomass due to various limitations of rainfall and other conditions in the plant photosynthetic process contracts the previous conclusions. Thus authors are making simplistic claims without considering the holistic impact of CO2 increasing concentrations on the biomes that are less studied at the molecular levels and at the experimental forest dynamics due to changing temperatures. Overall, I confirm that current level of understanding and justifications provided does not warrant the publication of this article.

---

## Short Comment (SC1) · 21 Jul 2020

**General comments**

The authors modified a dynamic-vegetation-model aDGVM then applied it to the South-Asia. After evaluating the simulation results under the historical climatic conditions, the modified model was forced by predicted climates and $CO_2$ trends, predicting major changes in geographical distribution of vegetation occurs by the end of the 21st century. A sensitivity test (i.e. comparing simulation results of four combinations of two $CO_2$ scenarios and two climate change scenarios), authors concluded that South Asia will likely to function as carbon sink during the 21st century due to the $CO_2$ fertilization effect. I evaluate that the manuscript is within the scope of the journal and it meets a basic scientific quality, however, authors need to address following items before publication.

**Major concerns**

(1) In the modification of aDGVM, a well-known functional relationships of leaves were introduced: SLA (specific leaf area) negatively correlates with Na (leaf nitrogen content per unit area), and Na positively correlates with Vcmax (maximum carbo-hydroxylation rate of Rubisco per unit leaf area). I should note that there is also a negative and strong correlation of SLA with leaf longevity (Wright et al. 2004), and actually, this correlation is much more intense than for the correlation of SLA with Na. Discounting the negative correlation between SLA with leaf longevity in the current model should favor higher SLA than for actual circumstances in nature. Author have to discuss how this discounting can skew the simulation results at least.

Wright, I. J., et al. (2004). "The worldwide leaf economics spectrum." Nature 428(6985): 821-827.

(2) Although the model was forced by various climatic variables, the manuscript only states influences of air temperature and precipitation. As authors themselves mentioned importance of VPD (vapor pressure deficit) on the transpiration rate in the manuscript (lines 406-409), other climatic variables controls the simulation. Accordingly, analysis and discussion how changes in other climatic variables influenced the results would be added. Besides, geographical distributions of all climatic variables, those are employed in the simulation, would be presented in the manuscript for both means of base-line-period and predicted trend during the 21st century.

(3) To quantify water use efficiency (WUE), authors scaled transpiration rate by leaf biomass (section 2.9). It's unusual. WUE is generally defined as carbon gain per unit water loss (i.e. photosynthesis rate per unit transpiration rate), because transpiration

can be regarded as inevitable water lose during $CO_2$ uptake through stomata for photosynthesis (Lambers et al., 1998). If authors use WUE of their definition, they need to clarify its underlining reason.

Lambers, H., F. S. Chapin, and T. L. Pons (1998), Plant Physiological Ecology, Springer, New York.

**Minor concerns**

Line 15. "$eCO_2$"
This term should not be used before its definition.

Lines 70-72
I could not understand this sentence.

Lines 74-75. "potentially disruptive effect of increasing $CO_2$ on natural vegetation"
It's a misleading phrase. As authors repeatedly mentioned in this manuscript, higher atmospheric $CO_2$ enhances photosynthesis rate and water-use-efficiency for $C_3$ plants, although no major influences would be happed for $C_4$ plants. Disruptive effects of higher $CO_2$ can be expected only if we consider other factors such as lower leaf-cooling-effect due to lower transpiration rate.

Lines 81-82, "resulting from environmental filtering applied to traits of modeled plant individuals"
I could not understand this phrase.

Lines 156-164
Values for parameters should be presented.

Line 184-185
Due to the large inter-annual variability of precipitation, I cannot see apparent climate trends in the figure S1. Because predicted trends in precipitation considerably differ among regions of the South Asia (Figure S2b), it would be meaningless to discuss predicted trend of average precipitation over South Asia.

Line 185. "Western Ghats"
Line 262. "Brahmaputra basin"
Will you provide approximate longitude and latitude of this region?

Lines 191-198. Section 2.5

Geographical distributions of soil types and elevation would be presented. A table of soil properties of each soil type is also required. Also, please add explanation how elevation controls the simulation.

Line 200. "four different scenarios"

Immediately after this term, an explanation would be needed that the four scenarios is the combination of two climate assumptions and two $CO_2$ assumptions.

Lines 224-225. "simulated vegetation stands (1 hectare)"

It means that the simulation unit of the aDGVM is 1 hectare? But, all forcing data and validation data were converted to 0.5 degree, and all simulation results were presented at the 0.5 degree resolution. Please add explanation.

Lines 230-231

Each tree can have multiple stem in the aDGVM? Need some explanation in the model description part of the supplemental material.

Lines 231-233

I could not understand this sentence. Please rewrite.

Line 271. "changes"

This word would be better to be replaced by "increasing trends".

Line 290. "grassland"

It would be better to be replaced by "grasslands", which contain both $C_3$ and $C_4$ grasslands.

Lines 335-336

Phrase "until 2090s" or "by 2090s" would be inserted somewhere in this sentence.

Line 399. "resulted"

It would be better to be replaced by "coincided with".

Tables S1 and S2

Units are missing.

Figure S5
No definition for the abbreviation "GRBM".

Figure S3, S4, S6, and S8
For convenience of readers, captions of these figures might be better to be replaced by "Same as the Fig * except RCP8.5".

**Typos**

Line 116
Year is missing for "Gillard et al."

Line 129. "Spack"
It would be replace by "$S_{pack}$". Unit for $S_{pack}$ should be also presented.

Line 171 "$C_4$ grasses"
"$C_3$ grasses"

Line 181. "$Wm^2$"
"$W/m^2$"

Line 314. "Ebiome"
It would be replace by "$E_{biome}$".

---

## Short Comment (SC2) · 21 Jul 2020

I should note that I posted my referee comments as "short comments" as I cannot post it as anonymous "referee comments".

---

## Referee Comment (RC3) · Hisashi Sato (Referee) · 22 Jul 2020

The editorial office checked the system and found out that I had two different user IDs which caused this problem. In addition to the account I normally use, I had different ID, which is connected to the referee role. Now, the problem was fixed and I submitted my referee report. So, please ignore this thread.

---

## Author Comment (AC1) · 14 Aug 2020

Dear Editor,

Review of the manuscript "Climate change and elevated CO2 favour forest over savanna under different future scenarios in South Asia.

The manuscripts read well and has the most relevance topic covered for the day given we are experiencing unprecedented climate change impacts and resultant pandemic. The aim to look into this south Asian scenario on forest biomass and changing forest cover over the region given the major of the population are dependent on natural resources for their livelihoods. No doubt these models developed point at the future direction and provide policy makers sufficient time to plan and execute appropriate policies aiming to conserve limited natural resources and protect the livelihoods of millions.

**Reply: Thank you for your appreciation and the positive feedback on the relevance of our work.**

The manuscript postulates that elevated CO2 levels improve the C3 function to enhance biomass productivity, which is quite simple to understand and model based on this assumption. However, at molecular level, the RUBP carboxylase has a different function based on the CO2 concentration, temperature and sunlight. Photosynthetic enzyme RUBISCO has dual function that fixes carbon when there is higher concentration of CO2 while at arid situations may act as RUBP oxygenase i.e., reduce carbon instead of fixing it. Thus, I request authors to go through the following articles which provide molecular basis of carbon fixation and how this might be helpful to model the scenarios under climate change.

1. James R Ehleringer and Thure E Cerling 2002, C3 and C4 Photosynthesis 2002 Volume 2, The Earth system: biological and ecological dimensions of global environmental change, pp 186–190 Edited by Professor Harold A Mooney and Dr Josep G Canadell in Encyclopedia of Global Environmental Change (ISBN 0-471-97796-9) Editor-in-Chief Ted Munn, John Wiley & Sons, Ltd, Chichester

https://d1wqtxts1xzle7.cloudfront.net/38426075/C3_ad_C4_photosynthesis.pdf?1439136433 =&response- content disposition=inline%3B+filename%3DC3_and_C4_Photosynthesis.pdf&Expires=1593404295 &Signature=aSHD6G 2uvohaB9ocTzzsdtH2Gg0pkJ3elFWyEmFKxx8GPH~hbRUY8GEXa- 3HwNWKtzR33dLW~hGJlb-WPoisOwxVaU227JcrqW8SRNTEQx3- wbiRVSw4hMBzhwY7heqUnffNwPy7avhdQnZQ1DFRR5DxBGvSP76vxWFmkxNTpDiR C4TP8KCRJM2YZZTH04a5LqXG Pair-Id=APKAJLOHF5GGSLRBV4ZA

2. Ian E. Woodrow and Joseph A. Berry 1988 ENZYMATIC REGULATION OF PHOTOSYNTHETIC CO2 FIXATION IN C3 PLANTS Ann. Rev. Plant Physiol. Plant Mol. Bioi. 1988. 39:533-94 https://www.annualreviews.org/doi/pdf/10.1146/annurev.pp.39.060188.002533

**Reply: Thank you for highlighting these two articles that provides detailed information on $CO_2$ fertilization under different environmental conditions. We will add these two references in our revised manuscript.**

**Thank you as well for pointing out your concerns about the effect of augmented photorespiration and its representation in the model at higher temperatures. We are well aware of how photosynthesis works and the dual nature of the ribulose-1, 5-bisphosphate-carboxylase-oxygenase enzyme. The photosynthesis implementation of Collatz et al. (1991, 1992) of the Farquhar photosynthesis scheme (Farquhar et al., 1980), combined with the Ball et al. (1987) implementation of stomatal conductance in aDGVM2 (as in many other DGVMs) accounts for these effects. The calculation of the $CO_2$ compensation point (gammastar) in this scheme depicts the dependency of carboxylation vs. oxygenation as a function of oxygen partial pressure and temperature (the latter via a Q10-function), and the $CO_2$ compensation point is then used further to determine Je (electron transport-limited photosynthesis, this also takes into account photosynthetically active radiation, i.e., PAR) and Jc ($CO_2$-concentration-limited photosynthesis, this also accounts for temperature-dependent Vcmax). Vcmax, the maximum carboxylation velocity in the implementation in the version of aDGVM2 used in this study, is temperature-dependent and reaches a peak around 37 °C for C3 plants (42 °C for C4 plants), and declines again at higher temperatures. This mimics the combined effect of decreasing enzyme activity due to the increased competitory binding of $O_2$ at higher temperatures and eventually enzyme degradation at very high temperatures.**

**We will add more details on this in the model description section of the manuscript to emphasize more clearly that the physiological basics and processes adequately represent $CO_2$ and temperature effects on primary production in aDGVM2.**

Furthermore, the manuscript under discussion line 349 (page 12) predicts expansion of woody forests compared to Savannas in mountainous regions. Similarly predicts that in Deccan plateau may change to evergreen forests from deciduous forests. This appears a very sweeping statement, given there is variation in rainfall associated with increasing temperatures Deccan region of the south Asia may experience floods and frequent droughts. It is well established that soils in Deccan region are low in nitrogen and other micro-nutrients, which support higher biomass production. I am not sure whether such physical attributes were considered while modelling the forest cover change. In fact, the regions of the deccan

plateau currently experience severe droughts which may cast uncertainty of the results obtained by the Authors. Under these situations the conclusions drawn seems to be quite simplistic without applying appropriate knowledge in the forestry sector.

**Reply: We agree that a DGVM run at a 0.5 ° spatial resolution cannot capture all the fine regional details required to provide in-depth detailed management advice to local forestry. There are other types of vegetation models that are better suited to conduct detailed studies at the local to regional scale. We are however confident that the results from DGVMs are nonetheless valuable because they can indicate more general trends of vegetation trajectories across larger spatial and temporal scales. We will make this more clear in discussion section 4.7.**

**We are also aware that the model results depend on the climate and soil input data provided as forcing parameters to the model. If extreme events such as droughts and torrential rainfalls are provided by the input data, the effects of such events on vegetation can be simulated. As we depend on climate simulation scenarios conducted with General Circulation Models (GCMs) to conduct simulations for future vegetation dynamics, the uncertainty of the climate model projections with respect to representation of future climate naturally also affects our simulation results. We tried to address this issue by conducting simulations for two alternative climate projections (RCP4.5 and RCP8.5) that reflect different possible climate futures in order to evaluate a range of possible vegetation states associated with these climate projections. Also in response to the review of Hitashi Sato, we now provide additional supplementary material (Figs. S1, S2) that illustrate spatial distribution and temporal trends of the climate input variables we used to drive the aDGVM2 simulations.**

**Nutrients and nutrient cycling are so far not included in aDGVM2, therefore such effects cannot be captured by the model. Where such limitations are known to occur, we expect aDGVM2 to potentially over-predict vegetation productivity. We will point this limitation out more strongly in the discussion section 4.7.**

**We would like to emphasize that the model does not predict transition of deciduous to evergreen forest in the Deccan region. Rather, it predicts that both savannas and woodlands in the Deccan region may transition to deciduous forest due to increase in tree biomass and canopy cover. A transition from deciduous to evergreen forests has been simulated for the mountainous regions, i.e., the Himalayas and the Western Ghats. We will rephrase the corresponding sentences in the manuscript to clarify this more explicitly and avoid misunderstandings.**

**We will add more details in the revised manuscript to clarify these points.**

The authors at page 13 line 376 explain aptly that decreased biomass due to various limitations of rainfall and other conditions in the plant photosynthetic process contracts the previous conclusions. Thus authors are making simplistic claims without considering the holistic impact of CO2 increasing concentrations on the biomes that are less studied at the molecular levels and at the experimental forest dynamics due to changing temperatures. Overall, I confirm that current level of understanding and justifications provided does not warrant the publication of this article.

**Reply:** We would like to highlight here that the raised concern is taken care of by the photosynthesis routine implemented in the aDGVM2. Like many other DGVMs, the aDGVM2 is a process-based model, which implies that it represents physiological, phenological and demographic processes that integrate from the leaf level to the plant level to the community or stand level, and from there to larger spatial scales. This entails that all the effects you claim are lacking in our study, with the exception of nutrient limitation effects, are actually represented by the explicit implementation of the responsible processes. Effects caused by changing atmospheric $CO_2$ concentrations and rising temperatures, including changes in carboxylation vs. oxygenation, are captured by the Collatz et al. (1991, 1992) implementation of the Farquhar photosynthesis model (Farquhar et al., 1980), as explained in our response to your first point. Effects of water limitation on stomatal conductance are represented by the Ball et al. (1987) implementation of stomatal conductance that ties photosynthesis to stomatal conductance via a diffusion-gradient definition.

We will gladly include the respective references in the model description part to highlight the detailed representation of the specific processes relevant to our study. The explicit strength of DGVMs lies in their ability to not only represents such relevant processes explicitly, but also to be able to scale them across a multitude of scales.

To make this more clear for readers who are not familiar with the workings of DGVMs, we will add some more information on this in the introduction section where we advertise why a DGVM is an appropriate tool to address our research questions.

On page 13 line 376 we referred to the prediction under fixed $CO_2$ scenarios, where the $CO_2$ fertilization does not happen and as a result biomass decreases, whereas biomass increases under $eCO_2$. The change in mean annual precipitation and mean annual temperature shown in Figure S2 supports our statement that $CO_2$ fertilization compensates the effect of high temperature and reduced precipitation in some areas. We will rephrase the corresponding sentence in the manuscript to avoid confusion.

**REFERENCES**
- Ball, J. T., Woodrow, I. E. and Berry, J. A.: A model predicting stomatal conductance and its contribution to the control of photosynthesis under different environmental conditions, in Progress in photosynthesis research, pp. 221–224, Springer., 1987.
- Collatz, G. J., Ball, J. T., Grivet, C. and Berry, J. A.: Physiological and environmental regulation of stomatal conductance, photosynthesis and transpiration: a model that includes a laminar boundary layer, Agric. For. Meteorol., 54(2–4), 107–136, 1991.
- Collatz, G. J., Ribas-Carbo, M. and Berry, J. A.: Coupled photosynthesis-stomatal conductance model for leaves of C4 plants, Funct. Plant Biol., 19(5), 519–538, 1992.
- Farquhar, G. D., von Caemmerer, S. von and Berry, J. A.: A biochemical model of photosynthetic CO2 assimilation in leaves of C3 species, Planta, 149(1), 78–90, 1980.

---

## Author Comment (AC2) · 14 Aug 2020

Dear Dr. Hisashi Sato,

Thank you very much for the time and effort you have dedicated to provide valuable feedback on our manuscript. We found your comments extremely helpful. We will incorporate changes to address your main concerns and hope you will find our replies to your comments on the manuscript satisfactory.

Our replies to the comments are highlighted in bold.

Best Regards,
Dushyant Kumar and co-authors

General comments
The authors modified a dynamic-vegetation-model aDGVM then applied it to the South-Asia. After evaluating the simulation results under the historical climatic conditions, the modified model was forced by predicted climates and CO2 trends, predicting major changes in geographical distribution of vegetation occurs by the end of the $21^{st}$ century. A sensitivity test (i.e. comparing simulation results of four combinations of two CO2 scenarios and two climate change scenarios), authors concluded that South Asia will likely to function as carbon sink during the $21^{st}$ century due to the CO2 fertilization effect. I evaluate that the manuscript is within the scope of the journal and it meets a basic scientific quality, however, authors need to address following items before publication.
**Reply: Thank you for the positive feedback.**

Major concerns
In the modification of aDGVM, a well-known functional relationships of leaves were introduced: SLA (specific leaf area) negatively correlates with Na (leaf nitrogen content per unit area), and Na positively correlates with Vcmax (maximum carbo-hydroxylation rate of Rubisco per unit leaf area). I should note that there is also a negative and strong correlation of SLA with leaf longevity (Wright et al. 2004), and actually, this correlation is much more intense than for the correlation of SLA with Na. Discounting the negative correlation between SLA with leaf longevity in the current model should favor higher SLA than for actual circumstances in nature. Author have to discuss how this discounting can skew the simulation results at least.
Wright, I. J., et al. (2004). "The worldwide leaf economics spectrum." Nature 428(6985): 821-827.

**Reply: We appreciate that you highlight this important aspect of the SLA-$N_a$ relationship. In aDGVM2, we are using the relation published by Sakschewski et al. (2015).**
**In aDGVM2, in addition to the SLA-$N_a$ relationship, we have also implemented the $A_{SLA}$ –LL trade-off in such a way that it affects the leaf turnover rates. Leaves with high $A_{SLA}$ have higher turnover rates (i.e, shorter leaf longevity) than leaves with low $A_{SLA}$. We will add this detail in the revised manuscript. The $A_{SLA}$ -LL trade-off implies that deciduous behavior is advantageous in dry regions because trees which do not invest much carbon into their leaves per unit dry mass (higher $A_{SLA}$) may shed them (lower LL) during the dry season without losing too much carbon. On the other hand, trees that invest more carbon into their leaves to enhance their structural stability (i.e., trees that make low-SLA leaves) have longer leaf turnover times and tend to emerge as**

**evergreen in aDGVM2, because being deciduous for them would be costly with respect to carbon use efficiency.**

**We will add these details to the text, together with the reference to the model description in the discussion part in section 4.1. We will also discuss the complete $A_{SLA}$ -LL-$N_a$-$V_{cmax}$ relationship and its impacts on vegetation state via trait trade-off. We will also include the $A_{SLA}$ -LL relation in the model description.**

Although the model was forced by various climatic variables, the manuscript only states influences of air temperature and precipitation. As authors themselves mentioned importance of VPD (vapor pressure deficit) on the transpiration rate in the manuscript (lines 406-409), other climatic variables controls the simulation. Accordingly, analysis and discussion how changes in other climatic variables influenced the results would be added. Besides, geographical distributions of all climatic variables, those are employed in the simulation, would be presented in the manuscript for both means of base-line- period and predicted trend during the 21$^{st}$ century.

**Reply: Thank you for highlighting your concern on the significance of other climate variables controlling the vegetation dynamics. VPD is estimated in the model using relative humidity and saturated vapor pressure. Due to stomatal closure, photosynthetic rates under soil water stress conditions decline in aDGVM2 when atmospheric VPD increases.**

**Other than precipitation and temperature, aDGVM2 uses relative humidity, downwelling short and long wave radiation, and near-surface wind speed. We agree with the reviewer's idea of presenting both mean and predicted trend of these variables during the 21$^{st}$ century and will include a revised figure in the Supplementary Material (Figure S2).**

**We will add more detail and elaborate on the impact of other climate variables on simulated results in the revised manuscript to make this clear.**

To quantify water use efficiency (WUE), authors scaled transpiration rate by leaf biomass (section 2.9). It's unusual. WUE is generally defined as carbon gain per unit water loss (i.e. photosynthesis rate per unit transpiration rate), because transpiration can be regarded as inevitable water lose during CO2 uptake through stomata for photosynthesis (Lambers et al., 1998). If authors use WUE of their definition, they need to clarify its underlining reason. Lambers, H., F. S. Chapin, and T. L. Pons (1998), Plant Physiological Ecology, Springer, New York.

**Reply: Thank you for providing the reference. In this study, we have not used WUE and instead decided to present the change in transpiration at biome level. As both biome area and total amount of leaf biomass per biome are subject to change over time, changes in absolute transpiration quantity can result from either of the two changes, or from changes in water supply, or from changes in WUE. We therefore normalized transpiration to biome-level leaf biomass to eliminate effects caused by change in biome area and leaf biomass per biome. The normalization makes such differences in transpiration at biome level more comparable and independent of biome attributes such as area covered by a respective biome and biome-level biomass.**

**Choosing to normalize transpiration to leaf biomass integrates over both increased WUE combined with soil water availability constraints, and in our opinion in our specific case is therefore more suitable to characterize overall change in water balance over time at biome level, as it not only indicates water used to produce new biomass (as GPP over transpiration would express), but also includes water required to sustain existing biomass.**

**We will elaborate more precisely why we decided to normalize transpiration to unit biomass per area rather than choosing to go with WUE in section 2.9 of the revised manuscript.**

Minor concerns
Line 15. "eCO2"
This term should not be used before its definition.
**Reply: Thank you for pointing it out. We will make the correction in the revised manuscript and put the definition with the first use of the term.**

Lines 70-72
I could not understand this sentence.

**Reply: We will rephrase the sentence to make it clear in the revised manuscript.**

**Previous: "They were further limited when using contemporary environmental conditions to pre-define bioclimatic limits of plant functional types (PFTs), and when using fixed eco-physiological parameters, for example to model carbon allocation".**

**Rephrased: "These studies were further limited by the fact that they all used fixed eco-physiological parameters and traits, e.g., fixed carbon allocation values to assign carbon to plant biomass pools, fixed SLA, as well as pre-defined bioclimatic limits that were derived from contemporary climatology in order to constrain the spatial distribution of plant functional types (PFTs) at global scale".**

Lines 74-75. "potentially disruptive effect of increasing CO2 on natural vegetation"
It's a misleading phrase. As authors repeatedly mentioned in this manuscript, higher atmospheric CO2 enhances photosynthesis rate and water-use-efficiency for C3 plants, although no major influences would be happed for C4 plants. Disruptive effects of higher CO2 can be expected only if we consider other factors such as lower leaf-cooling-effect due to lower transpiration rate.

**Reply: Yes, we agree that the phrasing is misleading. We thank the reviewer for pointing it out and will rephrase the sentence in the revised manuscript.**

Lines 81-82, "resulting from environmental filtering applied to traits of modeled plant individuals"
I could not understand this phrase.

**Reply: Here we meant that in aDGVM2, the implemented novel process of selection and trait inheritance assembles plant communities that are well-adapted to given biotic and abiotic conditions. A detailed description about the implementation is given in Langan, 2019 and Scheiter et al. (2013).**

**We will rephrase the sentence to make this more clear in the revised manuscript.**

Lines 156-164
Values for parameters should be presented.

**Reply: In the revised manuscript, we will include the parameter values that define temperature limitation of carboxylation, as well as other parameter values used for calculation of leaf temperature.**
**We would like to point out that the air density in aDGVM2 is not constant and is derived from atmospheric pressure (101.325 kPa) at sea level by scaling according to air temperature and elevation following Burman et al. (1987).**

Line 184-185
Due to the large inter-annual variability of precipitation, I cannot see apparent climate trends in the figure S1. Because predicted trends in precipitation considerably differ among regions of the South Asia (Figure S2b), it would be meaningless to discuss predicted trend of average precipitation over South Asia.

**Reply: We agree that there is no clear overall trend in mean annual precipitation, so we will rephrase the text in the revised manuscript.**

Line 185. "Western Ghats"
Line 262. "Brahmaputra basin"
Will you provide approximate longitude and latitude of this region?

**Reply: The Western Ghats are located between latitude 73°- 77° E and longitude 8°N - 21°N, and the Brahmaputra basin ís located between latitude 28°N - 34°N and longitude 90°E - 96.5°E. We will include the coordinates in the revised manuscript.**

Lines 191-198. Section 2.5
Geographical distributions of soil types and elevation would be presented. A table of soil properties of each soil type is also required. Also, please add explanation how elevation controls the simulation.

**Reply: We have used the same soil data that were used in Langan et al. (2017) and the soil properties are given in the Appendix S2, Table1 in Langan et al. (2017) (https://onlinelibrary.wiley.com/action/downloadSupplement?doi=10.1111%2Fjbi.13018&file=jbi13018-sup-0002-AppendixS2.pdf). In the model elevation is used to calculate the atmospheric pressure at different altitudes. This is required to derive the respective partial pressures of $O_2$ and $CO_2$, and to scale air densities at different altitude. The partial pressure of oxygen is used in the photosynthesis routine to estimate the $CO_2$ compensation point (see more detailed explanation on the topic of how photosynthesis is implemented in aDGVM2 in our replies to the comments provided by anonymous reviewer 1).**

**We will add the soil type and elevation maps along with a table for soil properties in the supplementary material, and we will add a brief explanation in the methods section why elevation is required for aDGVM2 simulations.**

Line 200. "four different scenarios"
Immediately after this term, an explanation would be needed that the four scenarios is the combination of two climate assumptions and two CO2 assumptions.

**Reply: Thank you for pointing it out. We will rephrase the sentence and include the suggestion in the revised manuscript.**

Lines 224-225. "simulated vegetation stands (1 hectare)"
It means that the simulation unit of the aDGVM is 1 hectare? But, all forcing data and validation data were converted to 0.5 degree, and all simulation results were presented at the 0.5 degree resolution. Please add explanation.

**Reply: In aDGVM2, a "1 hectare plot" is assumed to be representative for a grid cell and results from these representative stands are scaled up according to the grid cell size in hectares, i.e., in the current study to a grid cell resolution of $0.5° \times 0.5°$. In our simulation we used 3600 different individuals per hectare, and photosynthesis and other physiological processes are calculated separately for each individual. Therefore, simulations are already computationally expensive and time-costly when conducting them for representative stands of 1ha. Even with modern technical means, simulating the number of individuals required to fill an entire 0.5° x 0.5° grid cell would exceed available computational power and/or require extremely long simulation durations that are simply not feasible. The "representative 1 ha approach" is therefore a pragmatic approach to balance adequate representation of trait diversity among individuals against technical constraints.**

**We will add this explanation in the revised manuscript.**

Lines 230-231
Each tree can have multiple stem in the aDGVM? Need some explanation in the model description part of the supplemental material.

**Reply: Thank you for the suggestion. In aDGVM2, the stem number of individual woody plants is a dynamic trait. This trait has been included by Gaillard et al., (2018) in order to allow simulation of shrubs vs. trees based on a functional trade-off between augmented access to soil water resources vs. height growth. It simulates shrubs as multi-stemmed woody plants whereas all woody individuals with a stem number between one and three are defined as trees. The classification of individuals into these two categories is done a posteriori, based on the model results. Stem numbers in a woody plant are emerging based on water availability, light availability and fire activity and are based on a trade-off between rapid height growth in single-stemmed trees and augmented efficiency of water uptake in multi-stemmed shrubs due to higher sapwood area per unit of woody biomass.**

**We will add this description in the model description section in the supplementary material.**

Lines 231-233
I could not understand this sentence. Please rewrite.

**Reply: Simulated woody plants in aDGVM2 can be trees if stem number is 3 or less, or shrubs if stem number is 4 or more (please see preceding explanation). We used the canopy cover of woody plants (including both trees and shrubs) to categorize the biomes. The canopy cover of woody plants in combination with grass biomass was used to separate woodland and savanna biomes. Grid cells with tree canopy cover greater than shrub canopy cover (i.e., a tree-dominated system) and a tree canopy cover between 5% and 45%, combined with a grass biomass below 100 kg/ha, were classified as woodland. However, a grid cell with the same characteristics was classified as savanna where grass biomass exceeded 100 kg/ha (Kumar et al., 2020).**

**We will add more details in the revised manuscript to make our point clear.**

Line 271. "changes"
This word would be better to be replaced by "increasing trends".

**Reply: Thank you for the suggestion, we will make the suggested change.**

Line 290. "grassland"
It would be better to be replaced by "grasslands", which contain both C3 and C4 grasslands.

**Reply: Thank you for the suggestion, we will make the suggested change in the revised manuscript.**

Lines 335-336
Phrase "until 2090s" or "by 2090s" would be inserted somewhere in this sentence.

**Reply: We will add the suggestion in the revised manuscript. Thank you.**

Line 399. "resulted"
It would be better to be replaced by "coincided with".

**Reply: We agree with the suggestion and will rephrase the sentence in the revised manuscript.**

Tables S1 and S2
Units are missing.

**Reply: We will add the missing units to the values in table S1 and S2 in the revised supplementary material.**

Figure S5
No definition for the abbreviation "GRBM".

**Reply: It is an abbreviation for "grass biomass". We will add the non-abbreviated term in the figure caption in the revised supplementary material.**

Figure S3, S4, S6, and S8
For convenience of readers, captions of these figures might be better to be replaced by "Same as the Fig * except RCP8.5".

**Reply: Thank you for the suggestion. We will address this concern in the revised supplementary material.**

Typos
Line 116
Year is missing for "Gillard et al."

**Reply: Thanks for pointing it out. We will add the year in the revised manuscript.**

Line 129. "Spack"
It would be replace by "S$_{pack}$". Unit for S$_{pack}$ should be also presented.

**Reply: Thank you, we will add the suggestion in the revised manuscript.**

Line 171 "C4 grasses" "C3 grasses"
**Reply: Thank you, we will make the suggested correction in the revised manuscript.**

Line 181. "Wm$^2$" "W/m$^2$"

**Reply: We will make the suggested correction in the revised manuscript.**

Line 314. "Ebiome"
It would be replace by "E$_{biome}$".

**Reply: We will correct the mistake in the revised manuscript.**

**REFERENCE**
- Burman, R. D., Jensen, M. and Allen, R. G.: Thermodynamic factors in evapotranspiration, in Irrigation Systems for the 21st Century, pp. 140–148, ASCE, 1987.
- Gaillard, C., Langan, L., Pfeiffer, M., Kumar, D., Martens, C., Higgins, S. I. and Scheiter, S.: African shrub distribution emerges via height—Sapwood conductivity trade-off, J Biogeogr. 2018;00:1–12, doi:10.1111/jbi.13447, 2018.
- Kumar, D., Pfeiffer, M., Gaillard, C., Langan, L., Martens, C. and Scheiter, S.: Misinterpretation of Asian savannas as degraded forest can mislead management and conservation policy under climate change, Biol. Conserv., 241, 108293, 2020.
- Langan, L.: Holism in Plant Biogeography-Improving the Representation Of, and Interactions Between, the Biosphere, Hydrosphere, Atmosphere and Pedosphere, PhD Thesis, Johann Wolfgang Goethe-Universität Frankfurt am Main, 2019.
- Langan, L., Higgins, S. I. and Scheiter, S.: Climate-biomes, pedo-biomes or pyro-biomes: which world view explains the tropical forest–savanna boundary in South America?, J. Biogeogr., 44(10), 2319–2330, 2017.
- Sakschewski, B., Bloh, W., Boit, A., Rammig, A., Kattge, J., Poorter, L., Peñuelas, J. and Thonicke, K.: Leaf and stem economics spectra drive diversity of functional plant traits in a dynamic global vegetation model, Glob. Change Biol., 21(7), 2711–2725, 2015.
- Scheiter, S., Langan, L. and Higgins, S. I.: Next-generation dynamic global vegetation models: learning from community ecology, New Phytol., 198(3), 957–969, 2013.

---

## Author Comment (AC3) · 14 Aug 2020

This SC1 is identical to review RC2. We therefore only respond to review RC2 by Dr.Hisashi Sato.

---

## Author Comment (AC4) · 14 Aug 2020

This SC2 is identical to review RC2. We therefore only respond to review RC2 by Dr.Hisashi Sato.

---

## Author Comment (AC5) · 14 Aug 2020

This RC3 is identical to review RC2. We therefore only respond to review RC2 by Dr.Hisashi Sato.

---

## Author Response (AR1)

Dear Editor,

Please find the revised version of the manuscript. We have revised the manuscript and have carefully considered all the comments and suggestions by the referee. We provide point-by-point responses to all comments. In addition, we have also checked the manuscript for typos and reworded the text where necessary.

In the revised manuscript the addition of new texts are highlighted in blue underlined font and texts removed are highlighted in red font.

Our replies to the comments are highlighted in bold. We are looking forward to your decision,

Best regard,
Dushyant Kumar and co-authors

**Responses to Editor's comment**

Associate Editor Decision: Reconsider after major revisions (03 Sep 2020) by Eyal Rotenberg Comments to the Author: August 2020.

I would like to thanks the authors for the detailed response to the reviewers' comments. Outcome of the reviewers' recommendations leads to deiced for a major revision. When preparing your response please consider also the following points:

1. Please note that while addressing those comments you need to specifically answer them; the answer that it is included in the aDGVM2 model algorithm (e.g., page 4, the few first sentences is not enough.

**Reply: We have addressed the comments by providing a more detailed description of aDGVM2 and particularly of the algorithms used to calculate ecophysiological rates. Please see the response to the reviewer for more detail.**

2. Projected temperature change for the whole region provided in the manuscript, is within the range of 1 – 6 degrees. However, the forecast of precipitation changes is not given, although it is important information to assess the projected climatic changes in the region. If possible please provide figures with maps of current air temperature and precipitation and their projected changes for the whole region, as well as their differences, as in the panels of figure 1.

**Reply: In the revised version, we provide maps showing the current and projected changes until the 2050s and the 2090s of all climate variables derived from GCM (GFDL-ESM2M) used in the presented simulations. This includes maps for temperature and precipitation. See Fig. S2 in the supplementary materials.**

3. Figure 6 presents annual ET values ranged between 300 to 440 mm/a (0.8-1.2 mm/d), for the given MAP range. Given the region high precipitation range (800 to 1600 mm/a in that figure), those ET values seems very low and are not match the values of Figure 1. Please explain this discrepancy.

**Reply: The ET values presented in figure 1 represent the absolute value for specific grid cells, whereas the values used in Figure 6 are the spatial averages of daily ET per unit area over the entire study region (i.e., entire South Asia).**

These averages are compared with the spatial average of MAP over the region. This is the reason why the value of ET seems lower than corresponding MAP. We have mentioned this in Section 3.5 as well as in the Fig 6 caption.

4. Large number of papers predicts tree mortality due to future higher air temperatures and precipitation reduction (e.g., (Choat, Brodribb et al. 2018), (McDowell, Williams et al. 2016)), extension of the Earth dryland area over wetter regions (e.g., (Huang, Yu et al. 2016)) and many other studies show that the CO2 fertilization effect could not overcome the higher atmospheric water demand thus considerable WUE changes (e.g., Franks PJ,2014)). Although there are others opinions, naturally, the authors cannot ignore those effects in stating that (starting from line 346): "Under future conditions, the spatial distribution, extent and biomass of evergreen forests mostly remained at the current state, and evergreen forests were more resistant to climate change than deciduous forests. Expansion of deciduous forest into open biomes due to increasing woody cover resulted in significant loss of savanna area in the Deccan region under both RCPs with eCO2 by the end of the century. Transition from deciduous forests to evergreen forest was simulated for the mountain regions of South Asia."

Choat, B., et al. (2018). "Triggers of tree mortality under drought." Nature 558(7711): 531-539.

Huang, J. P., et al. (2016). "Accelerated dryland expansion under climate change." Nature Climate Change 6(2): 166-+.

McDowell, N. G., et al. (2016). "Multi-scale predictions of massive conifer mortality due to chronic temperature rise." Nature Climate Change 6(3): 295-300.

Franks PJ, Royer DL, Beerling DJ, Van de Water PK, Cantrill DJ, Barbour MM, Berry JA (2014) New constraints on atmospheric co2 concentration for the Phanerozoic. Geophys Res Lett 41: 4685–4694.

**Reply: Yes, we agree that other limitation such as drought, higher temperature and nutrient limitation can constraint or limit the effect of $CO_2$ fertilization and other related outcomes such as woody encroachment and phenology change and cannot be ignored. The main driver of the productivity in DGVMs is the increasing atmospheric $CO_2$ concentration (Zhu et al., 2017). Ecosystem-scale experiments indicate that nutrient constraints on plant functioning reduce the sensitivity of plant productivity to elevated $CO_2$ to such an extent that either the increase of productivity lasts only for few years (e.g. Norby et al., 2010) or no increase is observed (Ellsworth et al., 2017). The model version of aDGVM2 used for this study does not contain the nutrient cycle, which we have discussed these limitations in section 4.1 and 4.6.**

**To account for this uncertainty, we did $eCO_2$ and $fCO_2$ simulations to get the potential outcomes under the presence and absence of the effect to $CO_2$ fertilization at $eCO_2$.**

**We have added more details in Section 4.1 about the uncertainty due to other limiting factors on $CO_2$ fertilization effect on ecosystem.**

**REFERENCES**

- Norby, Richard J., Jeffrey M. Warren, Colleen M. Iversen, Belinda E. Medlyn, and Ross E. McMurtrie. "CO2 enhancement of forest productivity constrained by limited nitrogen availability." Proceedings of the National Academy of Sciences 107, no. 45 (2010): 19368-19373.

- Zhu, Zaichun, Shilong Piao, Yaoya Xu, Ana Bastos, Philippe Ciais, and Shushi Peng. "The effects of teleconnections on carbon fluxes of global terrestrial ecosystems." Geophysical Research Letters 44, no. 7 (2017): 3209-3218.

- Ellsworth, David S., Ian C. Anderson, Kristine Y. Crous, Julia Cooke, John E. Drake, Andrew N. Gherlenda, Teresa E. Gimeno et al. "Elevated CO 2 does not increase eucalypt forest productivity on a low-phosphorus soil." Nature Climate Change 7, no. 4 (2017): 279-282.

**Responses to RC2 by Anonymous Referee #1**

Dear Editor,
Review of the manuscript "Climate change and elevated CO2 favour forest over savanna under different future scenarios in South Asia. The manuscripts read well and has the most relevance topic covered for the day given we are experiencing unprecedented climate change impacts and resultant pandemic. The aim to look into this south Asian scenario on forest biomass and changing forest cover over the region given the major of the population are dependent on natural resources for their livelihoods. No doubt these models developed point at the future direction and provide policy makers sufficient time to plan and execute appropriate policies aiming to conserve limited natural resources and protect the livelihoods of millions.

**Reply: Thank you for your appreciation and the positive feedback on the relevance of our work.**

The manuscript postulates that elevated CO2 levels improve the C3 function to enhance biomass productivity, which is quite simple to understand and model based on this assumption. However, at molecular level, the RUBP carboxylase has a different function based on the CO2 concentration, temperature and sunlight. Photosynthetic enzyme RUBISCO has dual function that fixes carbon when there is higher concentration of CO2 while at arid situations may act as RUBP oxygenase i.e., reduce carbon instead of fixing it. Thus, I request authors to go through the following articles which provide molecular basis of carbon fixation and how this might be helpful to model the scenarios under climate change.

1. James R Ehleringer and Thure E Cerling 2002, C3 and C4 Photosynthesis 2002 Volume 2, The Earth system: biological and ecological dimensions of global environmental change, pp 186–190 Edited by Professor Harold A Mooney and Dr Josep G Canadell in Encyclopedia of Global Environmental Change (ISBN 0-471-97796-9) Editor-in-Chief Ted Munn, John Wiley Sons, Ltd, Chichester

```
https://d1wqtxts1xzle7.cloudfront.net/38426075/C3_ad_C4_photosynthesis.pdf?1
439136433=&response-content-disposition=inline%3B+filename%3DC3_and_C4_Photo
synthesis.pdf&Expires=1601907932&Signature=STAk6mLUGsJFRRBFNCtvCzzPw9oUkDX
I~1~d86lZ13fTtz5GqP-KH-jEnyrBqA5PRo54rWhZVc2PAR370kOCVOphKvteYqXT1bWzjWTlGr
5PeqlqS8KC1JASAU1OSs9z2ED8yL0v2Dt1Lhu5b3s0AnOC5cKvSPlOCOkewtBSQr~cnR8iyJw6M5
l0mT5-7lrhDVS2Oqvc452AdHCAGTJ5lqY9NctJMQ97JbFPHIO75zwE92f9qBIMzI9RLP-PXZi3xx
715LjobiytOoKzrzrKWPgALZA-yt7EwdhGfmKilnJAz3YY6RbPE2W01~eqSdfYk5mGaIXasD1vP
LOzzTdAJA_&Key-Pair-Id=APKAJLOHF5GGSLRBV4ZA
```

2. Ian E. Woodrow and Joseph A. Berry 1988 ENZYMATIC REGULATION OF PHOTO-SYNTHETIC CO2 FIXATION IN C3 PLANTS Ann. Rev. Plant Physiol. Plant Mol. Bioi. 1988. 39:533-94 `https://www.annualreviews.org/doi/pdf/10.1146/annurev.pp.39.06`
`0188.002533`

**Reply: Thank you for highlighting these two articles that provides detailed information on CO$_2$ fertilization under different environmental conditions. Thank you as well for pointing out your concerns about the effect of augmented photorespiration and its representation in the model at higher temperatures. We are well aware of how photosynthesis works and the dual nature of the ribulose-1, 5-bisphosphate-carboxylase-oxygenase enzyme. The photosynthesis implementation of Collatz et al. (1991, 1992) of the Farquhar photosynthesis scheme (Farquhar et al., 1980), combined with the Ball et al. (1987) implementation of stomatal conductance in aDGVM2 (as in many other DGVMs) accounts for**

these effects. The calculation of the $CO_2$ compensation point (gammastar) in this scheme depicts the dependency of carboxylation vs. oxygenation as a function of oxygen partial pressure and temperature (the latter via a Q10-function), and the $CO_2$ compensation point is then used further to determine Je (electron transport-limited photosynthesis, this also takes into account photosynthetically active radiation, i.e., PAR) and Jc ($CO_2$-concentration-limited photosynthesis, this also accounts for temperature-dependent Vcmax). Vcmax, the maximum carboxylation velocity in the implementation in the version of aDGVM2 used in this study, is temperature-dependent and reaches peaks around 37°C for $C_3$ plants and 42°C for $C_4$ plants, and declines again at higher temperatures. This mimics the combined effect of decreasing enzyme activity due to the increased competitory binding of $O_2$ at higher temperatures and eventually enzyme degradation at very high temperatures.

We have highlighted these aspects in first paragraph of section 2.2 and added details to emphasize more clearly that the physiological basics and processes adequately represent $CO_2$ and temperature effects on primary production in aDGVM2, in the supplementary material of the manuscript under model description.

Furthermore, the manuscript under discussion line 349 (page 12) predicts expansion of woody forests compared to Savannas in mountainous regions. Similarly predicts that in Deccan plateau may change to evergreen forests from deciduous forests. This appears a very sweeping statement, given there is variation in rainfall associated with increasing temperatures Deccan region of the south Asia may experience floods and frequent droughts. It is well established that soils in Deccan region are low in nitrogen and other micro-nutrients, which support higher biomass production. I am not sure whether such physical attributes were considered while modelling the forest cover change. In fact, the regions of the Deccan plateau currently experience severe droughts which may cast uncertainty of the results obtained by the Authors. Under these situations the conclusions drawn seems to be quite simplistic without applying appropriate knowledge in the forestry sector.

Reply: We agree that a DGVM run at a 0.5° spatial resolution cannot capture all the fine regional details required to provide in-depth detailed management advice to local forestry. There are other types of vegetation models that are better suited to conduct detailed studies at the local to regional scale. We are however confident that the results from DGVMs are nonetheless valuable because they can indicate more general trends of vegetation trajectories across larger spatial and temporal scales. We highlighted this concern in the discussion section 4.6.

We are also aware that the model results depend on the climate and soil input data provided as forcing parameters to the model. If extreme events such as droughts and torrential rainfalls are represented by the climate input data, the effects of such events on vegetation can be simulated. As we depend on climate simulation scenarios conducted with General Circulation Models (GCMs) to conduct simulations for future vegetation dynamics, the uncertainty of the climate model projections with respect to representation of future climate naturally also affects our simulation results. We addressed this issue by conducting simulations for two alternative climate projections (RCP4.5 and RCP8.5) that reflect different possible climate futures in order to evaluate a range of possible vegetation states associated with these climate projections. Also in response to the review

of Hisashi Sato, we now provide additional supplementary material (Figs. S1, S2) that illustrate spatial distribution of the climate input variables we used to drive the aDGVM2 simulations and change until 2099 from the baseline (2000-2009).

Nutrients and nutrient cycling are so far not included in aDGVM2, therefore such effects cannot be captured by the model. Where such limitations are known to occur, we expect aDGVM2 to potentially over-predict vegetation productivity. We have discussed this point in the discussion section 4.6.

We would like to emphasize that the model does not predict transition of deciduous to evergreen forest in the Deccan region. Rather, it predicts that both savannas and woodlands in the Deccan region may transition to deciduous forest due to increase in tree biomass and canopy cover. A transition from deciduous to evergreen forests has been simulated for the mountainous regions, i.e., the Himalayas and the Western Ghats.

We have added more details to the corresponding sentences in the section 4.1 of the revised manuscript to avoid the misunderstandings.

The authors at page 13 line 376 explain aptly that decreased biomass due to various limitations of rainfall and other conditions in the plant photosynthetic process contracts the previous conclusions. Thus authors are making simplistic claims without considering the holistic impact of CO2 increasing concentrations on the biomes that are less studied at the molecular levels and at the experimental forest dynamics due to changing temperatures. Overall, I confirm that current level of understanding and justifications provided does not warrant the publication of this article.

Reply: We would like to highlight here that the raised concern is taken care of by the photosynthesis routine implemented in the aDGVM2. Like many other DGVMs, the aDGVM2 is a process-based model, which implies that it represents physiological, phenological and demographic processes that integrate from the leaf level to the plant level to the community or stand level, and from there to larger spatial scales. This entails that all the mentioned by the referee, with the exception of nutrient limitation effects, are actually represented by the explicit implementation of the responsible processes. Effects caused by changing atmospheric $CO_2$ concentrations and rising temperatures, including changes in carboxylation vs. oxygenation, are captured by the Collatz et al. (1991, 1992) implementation of the Farquhar photosynthesis model (Farquhar et al., 1980), as explained in our response to your first point. Effects of water limitation on stomatal conductance are represented by the Ball et al. (1987) implementation of stomatal conductance that ties photosynthesis to stomatal conductance via a diffusion-gradient definition.

We have included the respective references in the model description part to highlight the detailed representation of the specific processes relevant to our study. The explicit strength of DGVMs lies in their ability to not only represents such relevant processes explicitly, but also to be able to scale them across a multitude of scales.

To make this clearer for readers who are not familiar with the workings of DGVMs, we have added some more information on this in the Model description

section in the supplementary material of the revised manuscript. On page 13 line 376 we referred to the prediction under fixed $CO_2$ scenarios, where the $CO_2$ fertilization does not happen and as a result biomass decreases, whereas biomass increases under $eCO_2$. The change in mean annual precipitation and mean annual temperature shown in Figure S2 supports our statement that $CO_2$ fertilization compensates the effect of high temperature and reduced precipitation in some areas. We have rephrased the corresponding sentence in the manuscript to avoid confusion.

**REFERENCES**

- Ball, J. T., Woodrow, I. E. and Berry, J. A.: A model predicting stomatal conductance and its contribution to the control of photosynthesis under different environmental conditions, in Progress in photosynthesis research, pp. 221–224, Springer., 1987.

- Collatz, G. J., Ball, J. T., Grivet, C. and Berry, J. A.: Physiological and environmental regulation of stomatal conductance, photosynthesis and transpiration: a model that includes a laminar boundary layer, Agric. For. Meteorol., 54(2–4), 107–136, 1991.

- Collatz, G. J., Ribas-Carbo, M. and Berry, J. A.: Coupled photosynthesis-stomatal conductance model for leaves of C4 plants, Funct. Plant Biol., 19(5), 519–538, 1992.

- Farquhar, G. D., von Caemmerer, S. von and Berry, J. A.: A biochemical model of photosynthetic CO2 assimilation in leaves of C3 species, Planta, 149(1), 78–90, 1980.

**Responses to RC2 by Dr. Hisashi Sato**

**General comments**

The authors modified a dynamic-vegetation-model aDGVM then applied it to the South-Asia. After evaluating the simulation results under the historical climatic conditions, the modified model was forced by predicted climates and CO2 trends, predicting major changes in geographical distribution of vegetation occurs by the end of the 21st century. A sensitivity test (i.e. comparing simulation results of four combinations of two CO2 scenarios and two climate change scenarios), authors concluded that South Asia will likely to function as carbon sink during the 21st century due to the CO2 fertilization effect. I evaluate that the manuscript is within the scope of the journal and it meets a basic scientific quality, however, authors need to address following items before publication.

**Reply: Thank you for the positive feedback.**

**Major concerns**

In the modification of aDGVM, a well-known functional relationships of leaves were introduced: SLA (specific leaf area) negatively correlates with Na (leaf nitrogen content per unit area), and Na positively correlates with Vcmax (maximum carbo-hydroxylation rate of Rubisco per unit leaf area). I should note that there is also a negative and strong correlation of SLA with leaf longevity (Wright et al. 2004), and actually, this correlation is much more intense than for the correlation of SLA with Na. Discounting the negative correlation between SLA with leaf longevity in the current model should favor higher SLA than for actual circumstances in nature. Author have to discuss how this discounting can skew the simulation results at least. Wright, I. J., et al. (2004). "The worldwide leaf economics spectrum." Nature 428(6985): 821-827.

**Reply: We appreciate that you highlight this important aspect of leaf economy. The mentioned relationship between SLA and leaf longevity is already included in aDGVM2 (see detailed model description in Langan et al. 2017), and the SLA-Na trade-off affects the leaf turnover rates. Leaves with high SLA have higher turnover rates (i.e, shorter leaf longevity) than leaves with low SLA. The SLA -LL trade-off implies that deciduous behavior is advantageous in dry regions because trees which do not invest much carbon into their leaves per unit dry mass (higher SLA) may shed them (lower LL) during the dry season without losing too much carbon. On the other hand, trees that invest more carbon into their leaves to enhance their structural stability (i.e., trees with low-SLA leaves) have longer leaf turnover times and tend to emerge as evergreen in aDGVM2. Being deciduous would be costly with respect to carbon use efficiency under such condition.**

**We have added these details to the text in Section 2.2 under sub-section (b) Carboxylation rate, in the revised manuscript. We have discussed the complete SLA -LL-Na-Vcmax relationship and its impacts on vegetation state via trait trade-off in the discussion part in section 4.1.**

Although the model was forced by various climatic variables, the manuscript only states influences of air temperature and precipitation. As authors themselves mentioned importance of VPD (vapor pressure deficit) on the transpiration rate in the manuscript (lines 406-409), other climatic variables controls the simulation. Accordingly, analysis and discussion how changes in other climatic variables influenced the results would be added. Besides, geographical distributions of all climatic variables, those are employed in the simulation, would be presented in the manuscript for both means of base-line- period and predicted trend during

the 21st century.

Reply: Thank you for highlighting your concern on the significance of other climate variables controlling the vegetation dynamics. In aDGVM2, VPD is calculated using relative humidity (one of the climate forcing) and saturated vapor pressure. Due to stomatal closure, photosynthetic rates under soil water stress conditions decline in aDGVM2 when atmospheric VPD increases.

Other than precipitation and temperature, aDGVM2 uses relative humidity, downwelling short and long wave radiation, and near-surface wind speed as climate forcing derived from from GCM (GFDL-ESM2M). We agree with the reviewer's idea of presenting both mean and predicted trend of these variables during the 21st century and included a revised figure in the Supplementary Material (Figure S2).

We have added details on the impact of projected climate change on simulated results in the revised manuscript in discussion sections from section 4.1, 4.2, 2.3 and 4.4

To quantify water use efficiency (WUE), authors scaled transpiration rate by leaf biomass (section 2.9). It's unusual. WUE is generally defined as carbon gain per unit water loss (i.e. photosynthesis rate per unit transpiration rate), because transpiration can be regarded as inevitable water lose during CO2 uptake through stomata for photosynthesis (Lambers et al., 1998). If authors use WUE of their definition, they need to clarify its underlining reason. Lambers, H., F. S. Chapin, and T. L. Pons (1998), Plant Physiological Ecology, Springer, New York.

Reply: Thank you for providing the reference. In this study, we have not used WUE. Instead we decided to present the change in transpiration at biome level. As both biome area and total amount of leaf biomass per biome are subject to change over time, changes in absolute transpiration quantity can result from area or biomass changes, or from changes in water supply, or from changes in WUE. We therefore normalized transpiration to biome-level leaf biomass to eliminate effects caused by change in biome area and leaf biomass per biome. The normalization makes such differences in transpiration at biome level more comparable and independent of biome attributes such as area covered by a respective biome and biome-level biomass.

Choosing to normalize transpiration to leaf biomass integrates over both increased WUE combined with soil water availability constraints. In our opinion, in our specific case is therefore more suitable to characterize overall change in water balance over time at biome level, as it not only indicates water used to produce new biomass (as GPP over transpiration would express), but also includes water required to sustain existing biomass.

We elaborated more precisely why we decided to normalize transpiration to unit biomass per area rather than choosing to go with WUE in section 2.7 of the revised manuscript.

Minor concerns

Line 15. "eCO2" This term should not be used before its definition. **Reply: Thank you for pointing it out. We made the correction in the revised manuscript and put**

the definition with the first use of the term.

Lines 70-72 I could not understand this sentence.

**Reply: We rephrased the sentence to make it clear in the revised manuscript.**

**Previous: "They were further limited when using contemporary environmental conditions to pre-define bioclimatic limits of plant functional types (PFTs), and when using fixed eco-physiological parameters, for example to model carbon allocation".**

**Rephrased:** *"These studies were further limited by the utilization of models with fixed eco-physiological parameters and traits e.g., fixed carbon allocation values to assign carbon to plant biomass pools, fixed specific leaf area (SLA), as well as pre-defined bioclimatic limits that were derived from contemporary climatology in order to constrain the spatial distribution of plant functional types (PFTs)".*

Lines 74-75. "potentially disruptive effect of increasing CO2 on natural vegetation" It's a misleading phrase. As authors repeatedly mentioned in this manuscript, higher atmospheric CO2 enhances photosynthesis rate and water-use-efficiency for C3 plants, although no major influences would be happed for C4 plants. Disruptive effects of higher CO2 can be expected only if we consider other factors such as lower leaf-cooling-effect due to lower transpiration rate.

**Reply: Yes, we agree that the phrasing is misleading. We thank the reviewer for pointing it out and rephrased the sentence in the revised manuscript.The modified sentence is**

*"While some global-scale studies have investigated the potentially disruptive effect of increasing CO2 on natural vegetation, carbon sequestration and biome boundaries (e.g. Hickler et al., 2006; Sato et al., 2007; Smith et al., 2013), detailed modeling studies focusing explicitly on different biomes in South Asia have not been conducted."*

Lines 81-82, "resulting from environmental filtering applied to traits of modeled plant individuals" I could not understand this phrase.

**Reply: Here we meant that in aDGVM2, the implemented novel process of selection and trait inheritance assembles plant communities that are well-adapted to given biotic and abiotic conditions. A detailed description about the implementation is given in Langan, 2019 and Scheiter et al. (2013).**

**We rephrased the sentence to make this clearer in the revised manuscript. The revised sentences is**

*"To address the knowledge gaps in existing studies, we used the aDGVM2 (adaptive dynamic global vegetation model version 2), an individual- and trait-based vegetation model that combines elements of traditional DGVMs (Prentice et al., 2007) with newly implemented approaches for selection and trait filtering. In aDGVM2, environmental conditions used for plant with trait value combinations that make them successful under these conditions. Therefore, plant communities that are adapted to site-specific environmental conditions dynamically assembles and emerges as a reaction to the environmental forcing*

*(Langan et al., 2017; Scheiter et al., 2013)."*

Lines 156-164 Values for parameters should be presented.

**Reply: In the revised manuscript section 2.2, we included the parameter values that define temperature limitation of carboxylation, as well as other parameter values used for calculation of leaf temperature. We would like to point out that the air density in aDGVM2 is not constant and is derived from atmospheric pressure (101.325 kPa) at sea level by scaling according to air temperature and elevation following Burman et al. (1987).**

Line 184-185 Due to the large inter-annual variability of precipitation, I cannot see apparent climate trends in the figure S1. Because predicted trends in precipitation considerably differ among regions of the South Asia (Figure S2b), it would be meaningless to discuss predicted trend of average precipitation over South Asia.

**Reply: We agree that there is no clear overall trend in mean annual precipitation, so we rephrased the text in the revised manuscript and have time-series of MAP shown in Fig S1 which shows that there is large inter- annual variability and no clear trend.**

Line 185. "Western Ghats" Line 262. "Brahmaputra basin" Will you provide approximate longitude and latitude of this region?

**Reply: The Western Ghats are located between latitude 73°- 77° E and longitude 8°N - 21°N, and the Brahmaputra basin is located between latitude 28°N - 34°N and longitude 90°E - 96.5°E. We included the coordinates in the revised manuscript.**

Lines 191-198. Section 2.5 Geographical distributions of soil types and elevation would be presented. A table of soil properties of each soil type is also required. Also, please add explanation how elevation controls the simulation.

**Reply: We have used the same soil data that were used in Langan et al. (2017) and the soil properties are given in the Appendix S2, Table1 in Langan et al. (2017) (`https://onlinelibrary.wiley.com/action/downloadSupplement?doi=10.1111 %2Fjbi.13018&file=jbi13018-sup-0002-AppendixS2.pdf`). In the model elevation is used to calculate the atmospheric pressure at different altitudes. This is required to derive the respective partial pressures of $O_2$ and $CO_2$, and to scale air densities at different altitude. The partial pressure of oxygen is used in the photosynthesis routine to estimate the $CO_2$ compensation point (see more detailed explanation on the topic of how photosynthesis is implemented in aDGVM2 in our replies to the comments provided by anonymous reviewer 1).**

**We added the soil type and elevation maps along with a table for soil properties in the supplementary material (Fig. S3), and we added a brief explanation in the methods (Section 2.3.2) why elevation is required for aDGVM2 simulations.**

Line 200. "four different scenarios" Immediately after this term, an explanation would be needed that the four scenarios is the combination of two climate assumptions and two CO2 assumptions.

**Reply: Thank you for pointing it out. We rephrased the sentence and included**

the suggestion in the revised manuscript.

*"To understand how climate change and $CO_2$ fertilization interact to influence the future vegetation state in South Asia, we simulated all combination of two climate scenarios (RCP4.5 and RCP8.5) and two $CO_2$ scenarios ($CO_2$ fertilization enabled or disabled, four scenarios in total)."*

Lines 224-225. "simulated vegetation stands (1 hectare)" It means that the simulation unit of the aDGVM is 1 hectare? But, all forcing data and validation data were converted to 0.5 degree, and all simulation results were presented at the 0.5 degree resolution. Please add explanation.

Reply: In aDGVM2, a "1 hectare plot" is assumed to be representative for a grid cell. Results from these representative stands are scaled up to the grid cell size i.e., in the current study to a grid cell resolution of 0.5° × 0.5° assuming that the vegetation in this grid cell is homogeneous. The aDGVM2 simulations are computationally expensive and time-costly when conducting them for representative stands of 1ha. Even with computer clusters that available for our simulations, simulating the number of individuals required to fill an entire 0.5° x 0.5° grid cell would exceed available computational power and/or require extremely long simulation durations that are simply not feasible. For the same reason, we did not conduct replicate simulation runs for each grid cell. The "representative 1 ha approach" is therefore a pragmatic approach to balance adequate representation of trait diversity among individuals against technical constraints.

We have added explanation in the revised manuscript in section 2.4 under "Model simulation protocol".

Lines 230-231 Each tree can have multiple stem in the aDGVM? Need some explanation in the model description part of the supplemental material.

Reply: Thank you for the suggestion. In aDGVM2, the stem number of individual woody plants is a dynamic trait. This trait has been included by Gaillard et al., (2018) in order to allow simulation of shrubs vs. trees based on a functional trade-off between efficient access to soil water resources and water uptake vs. rapid height growth. It simulates shrubs as multi-stemmed woody plants whereas all woody individuals with a stem number between one and three are defined as trees. The classification of individuals into these two categories is done a posteriori, based on the model results. Stem numbers in a woody plant are emerging based on water availability, light availability and fire activity as discussed in Gaillard et al. (2018).

We added this detail in the model description section in the supplementary material.

Lines 231-233 I could not understand this sentence. Please rewrite.

Reply: We rephrased the sentence in the revised manuscript to make our point clear in section 2.6 and added details in the supplementary material on shrubs.

*"The canopy cover of woody plants and grass biomass were used to separate woodland and savanna biomes. Grid cells with tree canopy cover greater than shrub canopy cover, tree canopy cover between 5% and 45%, and grass biomass*

*below 100 kg/ha, were classified as woodland. Grids cells with the same woody cover characteristics but grass biomass higher than 100 kg/ha were classified as savanna."*

**We rephrased the sentence in the revised manuscript to make our point clear and added details in the supplementary material on shrubs.**

Line 271. "changes" This word would be better to be replaced by "increasing trends".

**Reply: Thank you for the suggestion, we made the suggested change.**

Line 290. "grassland" It would be better to be replaced by "grasslands", which contain both C3 and C4 grasslands.

**Reply: Thank you for the suggestion, we made the suggested change in the revised manuscript.**

Lines 335-336 Phrase "until 2090s" or "by 2090s" would be inserted somewhere in this sentence.

**Reply: We added the suggestion in the revised manuscript. Thank you.**

Line 399. "resulted" It would be better to be replaced by "coincided with".

**Reply: We agree with the suggestion and rephrased the sentence in the revised manuscript.**

Tables S1 and S2 Units are missing.

**Reply: We added the units to the values of parameters in table S1 and S2 in the revised supplementary material. Values without specified units are fraction or unit less.**

Figure S5 No definition for the abbreviation "GRBM".

**Reply: It is an abbreviation for "grass biomass". We added the non-abbreviated term in the figure caption in the revised supplementary material.**

Figure S3, S4, S6, and S8 For convenience of readers, captions of these figures might be better to be replaced by "Same as the Fig * except RCP8.5".

**Reply: Thank you for the suggestion. But we feel that it would be easier to follow the figures if details are repeated in captions.**

Typos Line 116 Year is missing for "Gillard et al."

**Reply: Thanks for pointing it out. We added the year in the revised manuscript.**

Line 129. "Spack" It would be replace by "Spack". Unit for Spack should be also presented.

**Reply: Thank you, we added the suggestion in the revised manuscript.**

Line 171 "C4 grasses" "C3 grasses" **Reply: Thank you, we made the suggested correction in the revised manuscript.**

Line 181. "Wm2" "W/m2"

**Reply: We made the suggested correction in the revised manuscript.**

Line 314. "Ebiome" It would be replace by "Ebiome".

**Reply: We corrected the mistake in the revised manuscript.**

**REFERENCE**

- Burman, R. D., Jensen, M. and Allen, R. G.: Thermodynamic factors in evapotranspiration, in Irrigation Systems for the 21st Century, pp. 140–148, ASCE, 1987.

- Gaillard, C., Langan, L., Pfeiffer, M., Kumar, D., Martens, C., Higgins, S. I. and Scheiter, S.: African shrub distribution emerges via height—Sapwood conductivity trade-off, J Biogeogr. 2018;00:1–12, doi:10.1111/jbi.13447, 2018.

- Kumar, D., Pfeiffer, M., Gaillard, C., Langan, L., Martens, C. and Scheiter, S.: Misinterpretation of Asian savannas as degraded forest can mislead management and conservation policy under climate change, Biol. Conserv., 241, 108293, 2020.

- Langan, L., Higgins, S. I. and Scheiter, S.: Climate-biomes, pedo-biomes or pyro-biomes: which world view explains the tropical forest–savanna boundary in South America?, J. Biogeogr., 44(10), 2319–2330, 2017.

- Prentice, I. Colin, Alberte Bondeau, Wolfgang Cramer, Sandy P. Harrison, Thomas Hickler, Wolfgang Lucht, Stephen Sitch, Ben Smith, and Martin T. Sykes. "Dynamic global vegetation modeling: quantifying terrestrial ecosystem responses to large-scale environmental change." In Terrestrial ecosystems in a changing world, pp. 175-192. Springer, Berlin, Heidelberg, 2007.

- Scheiter, S., Langan, L. and Higgins, S. I.: Next-generation dynamic global vegetation models: learning from community ecology, New Phytol., 198(3), 957–969, 2013.

---

## Author Response (AR2)

Dear Dr. Eyal Rotenberg,

Please find the revised version of the manuscript. We have revised the manuscript and have carefully provided more evidence to support our model results.

In the revised manuscript the addition of new texts are highlighted in bold black font.

Our replies to the comments are highlighted in bold. We are looking forward to your decision,

Best regards,
Dushyant Kumar and co-authors

**Responses to Editor's comment**

Comments to the Author:
Dear Authors,

Thank you a lot for resubmitting your work and your efforts to address the reviewers' comments. Considering the amount of reviewers comments, their profoundness, their importance for the holistic view of the manuscript and your extensive replies, I find the revisions by the reviewers a necessarily step forward.

**Reply: We thank you for finding our revision promising. We have addressed both concerns on temperature-$CO_2$ sensitivity for biomass, and model prediction for phenology change. We have revised the manuscript and have added more details and evidence to support our model results.**

As already mentioned in my previous stage comments, many of the replies to comments you are considering that those effects included in the aDGVM2 model (e.g., temperature sensitivity) and that in current submission it is better described how the model treat those comments. I would suggest to provide further evidences substantiating the way of modelling of these effects, based, as much as possible, on field observations or others models for this or others regions.

**Reply: We have added the following paragraphs to section 4.2:**

**"The sensitivity of biomass to temperature and $CO_2$ change has been investigated in many studies (Norby and Luo, 2004; Jian et al., 2019; Sperry et al, 2017). A meta-analysis by Lin et al. (2010) showed that warming significantly increased biomass by 12.3% (with a 95% confidence interval of 8.4–16.3%) across all the terrestrial plants included. This observation is consistent with our model results. Biomass showed a positive relation with MAT which did not change with mean annual precipitation or experimental duration or $CO_2$ enrichment (Lin et al., 2010). These findings are also supported by previous studies by Rustad et al. (2001), Dormann Woodin (2002) and Walker et al. (2006) which have revealed that warming generally increases terrestrial plant biomass, indicating enhanced terrestrial carbon uptake via plant growth.**

**Previous modeling studies using Biome-BGC (Running and Hunt, 1993), Cen-**

tury (Parton et al., 1993), and TEM (Tian et al., 1999) have shown an increase in productivity when both climate change and $CO_2$ effects were considered. However, the increase was smaller when only climate change effects were considered and both Biome-BGC and TEM suggest that without $CO_2$ fertilization, average productivity would decline relative to current annual average as shown by our result (Fig. 6d)."

An example for such comment reply: Reviewer 1 stated that: "... the manuscript under discussion, line 349 (page 12), predicts the expansion of woody forests compared to Savannas in mountainous regions. Similarly predicts that in Deccan plateau may change to evergreen forests from deciduous forests" and he continue "... these situations and the conclusions drawn seem to be quite simplistic without applying appropriate knowledge in the forestry sector." Your reply is that: "A transition from deciduous to evergreen forests has been simulated for the mountainous regions, i.e., the Himalayas and the Western Ghats." Can you provide any observations/model results others than your model to support your claim?

Warm regards and wish you healthy days in those scary period, Eyal

**Reply: We have added the following paragraphs to the discussion in section 4.1:**

"**Phenology change as a result of climate change has already been observed (Buitenwerf, Rose Higgins, 2015; Cleland et al, 2007). In Scheiter et al. (2020), we showed that climate change supports transitions to tall evergreen vegetation in tropical Asia and found increases in the abundance of evergreen plants and decreases in the abundance of deciduous plants in mainland Southeast Asia, central India, and Pakistan. This relative advantage of evergreen plants over deciduous plants under elevated $CO_2$ in aDGVM2 can be explained by the fact that increased intrinsic water use efficiency under $eCO_2$ in evergreen plants are higher than in deciduous plants as demonstrated by Soh et al (2019).**

**Previous modeling studies also support aDGVM2 result showing transitions from deciduous to evergreen vegetation. With the BIOME4 model, Ravindranath et al. (2006) simulated the response of forest to SRES A2 and B2 scenarios and reported similar changes toward evergreen phenology. A study by Chaturvedi et al. (2011) using the IBIS model also predicted transitions toward evergreen forest.**

**Woody encroachment in many ecosystems is attributed to rising $CO_2$ and this is supported by studies based on both field observations (e.g., FACE experiments) and satellite data (Brienen et al., 2015; Archer et al., 2017; Stevens et al., 2016; Piao et al., 2006; Schimel et al., 2015). The aDGVM2 also supports these findings i.e., increasing canopy cover and woody biomass under the $eCO_2$ condition and agrees with the reported greening trend in South Asia during the last three decades (Wang et al., 2017)."**

**References**

- Song, Jian, Shiqiang Wan, Shilong Piao, Alan K. Knapp, Aimée T. Classen, Sara Vicca, Philippe Ciais et al. "A meta-analysis of 1,119 manipulative experiments on terrestrial carbon-cycling responses to global change." Nature ecology evolution 3, no. 9 (2019): 1309-1320.

- Sperry, John S., Martin D. Venturas, Henry N. Todd, Anna T. Trugman, William RL

Anderegg, Yujie Wang, and Xiaonan Tai. "The impact of rising CO2 and acclimation on the response of US forests to global warming." Proceedings of the National Academy of Sciences 116, no. 51 (2019): 25734-25744.

- Stevens, Nicola, B. F. N. Erasmus, S. Archibald, and W. J. Bond. "Woody encroachment over 70 years in South African savannahs: overgrazing, global change or extinction aftershock?." Philosophical Transactions of the Royal Society B: Biological Sciences 371, no. 1703 (2016): 20150437.

- Archer, Steven R., Erik M. Andersen, Katharine I. Predick, Susanne Schwinning, Robert J. Steidl, and Steven R. Woods. "Woody Plant Encroachment: Causes and Consequences." In Rangeland Systems, pp. 25-84. Springer, Cham, 2017.

- Buitenwerf, R., Rose, L., Higgins, S. I. (2015). Three decades of multi-dimensional change in global leaf phenology. Nature Climate Change, 5, 364–368. https://doi.org/10.1038/nclin

- Cleland, E. E., Chuine, I., Menzel, A., Mooney, H. A., Schwartz, M. D. (2007). Shifting plant phenology in response to global change. Trends in Ecology Evolution, 22, 357–365. https://doi.org/10.1016/j.tree.2007.04.003

- Soh, W. K., Yiotis, C., Murray, M., Parnell, A., Wright, I. J., Spicer, R. A., ... McElwain, J. C. (2019). Rising CO2 drives divergence in water use efficiency of evergreen and deciduous plants. Science Advances, 5, eaax7906. https://doi.org/10.1126/sciadv.aax7906

- Ravindranath, N. H., Joshi, N. V., Sukumar, R., Saxena, A. (2006). Impact of climate change on forests in India. Current Science, 90, 354–361.

- Chaturvedi, R. K., Gopalakrishnan, R., Jayaraman, M., Bala, G., Joshi, N. V., Sukumar, R., Ravindranath, N. H. (2011). Impact of climate change on Indian forests: A dynamic vegetation modeling approach. Mitigation and Adaptation Strategies for Global Change, 16, 119–142. https://doi.org/10.1007/s11027-010-9257-7

- Lin, Delu, Jianyang Xia, and Shiqiang Wan. "Climate warming and biomass accumulation of terrestrial plants: a meta-analysis." New Phytologist 188, no. 1 (2010): 187-198. https://doi.org/10.1111/j.1469-8137.2010.03347.x

- Norby, Richard J., and Yiqi Luo. "Evaluating ecosystem responses to rising atmospheric CO2 and global warming in a multi-factor world." New phytologist 162, no. 2 (2004): 281-293. https://doi.org/10.1111/j.1469-8137.2004.01047.x

- Running SW, Hunt ER Jr. 1993. Generalization of a forest ecosystem process model for other biomes, BIOME-BGC, and an application for global-scale models. In: Ehleringer JR, Field C, eds. Scaling processes between leaf and landscape levels. San Diego, CA, USA: Academic Press, 141–158.

- Tian H, Melillo JM, Kicklighter DW, McGuire AD, Helfrich J. 1999. The sensitivity of terrestrial carbon storage to historical climate variability and atmospheric CO2 in the United States. Tellus 51B: 414–452

- Parton WJ, Scurlock JMO, Ojima DS, Gilmanov TG, Scholes RJ, Schimel DS, Kirchner T, Menaut JC, Seastedt T, Moya EG, Kamnalrut A, Kinyamario JI. 1993. Observations and modeling of biomass and soil organic-matter dynamics for the grassland biome worldwide. Global Biogeochemical Cycles 7: 785–809.

---

## Author Response (AR3)

Dear Editor, dear Reviewers,

Thank you for the positive feedback, and the opportunity to resubmit a revised version of the manuscript. We revised the manuscript to address all comments and responded to all comments point by point.

Our replies to the comments and corresponding changes in the manuscript (manuscript with track change) are highlighted in bold.

We look forward to your decision.

Best Regards,
Dushyant Kumar and Co-authors.

**Comments by referee #2:**

"Thanks for the opportunity for reviewing the manuscript "Climate change and elevated $CO_2$ favor forest over savanna under different future scenarios in South Asia" by Kumar et al. This study uses aDGVM2 to simulate vegetation dynamics in South Asia under RCP4.5 and RCP8.5 projected by GFDL-ESM2M climate data for the period 1950 to 2099. This is entirely a modeling study, but the method description was detailed and the limitation of relying on modeling was well recognized.

**Reply: Thank you for your time in reviewing our manuscript.**
The writing was clear overall with sporadic grammar issues. I have the following major concerns.
1) Similar types of study have been done many times in the literature to examine climate and $CO_2$ effects on vegetation, through well-known mechanisms of drought and $CO_2$ fertilization. It would be useful for the authors to think about and explicitly describe what is novel for this particular study.

Reply: **Thank you for pointing this out. Indeed there are many studies available for the region, however, most of those studies are either global scale and have represented the region with few vegetation types with fixed trait variability or focused mainly on forest neglecting the savanna biomes. The novelty of our study is that this is the first study where we have used a fully trait based vegetation model i.e., aDGVM2, which is capable of considering trait variability and how plants adapt to changing environmental conditions. We also considered the long neglected and misrepresented savanna biomes. We also conducted simulation with/without $CO_2$ to investigate the role of CO2 fertilization, however the impact of climate change and rising $CO_2$ effects are uncertain.**

**We have highlighted these explanation in the revised manuscript in bold in line 70-78.**

2) Authors used MODIS ET product to validate model predictions. MODIS is per unit ground area based, while the authors converted model predictions into per biomass based. It seems to me

these two have different meanings and should not be compared directly. Further, MODIS ET also provides temporal sequence, but the authors seemed to have only validated the spatial pattern of model predicted ET. What about temporal dynamics?

Reply: **For model-data comparison (Fig. 1 and S3) , we used a 10 year average of MODIS ET and compared it to a 10 year average of model simulated ET (2000-2009). The modelled ET used for data-model comparison has also same unit as MODIS ET i.e., $mm/m^2/year$. For performing comparative analysis of biome level ET under different scenarios for different periods (Table 3), we estimated the ET per unit biomass for respective biome i.e., mm/kg/year. We normalized ET to biome-level leaf biomass to eliminate effects caused by change in biome area and leaf biomass per biome and make it more comparable and independent of biome attributes such as area covered by a respective biome and biome-level biomass.**

**For clarification, we have highlighted the units used for data-model comparison of ET in line 240-242 and biome level ET in line 282 in bold.**

**In our study we did not focus on the temporal dynamics of the ET and only used the decadal average for comparison. However, we agree that it would be very interesting to look at temporal dynamics to assess the performance of aDGVM2 in capturing the seasonality of ET in the ecosystem.**

3) It seems aDGVM2 predicts biome changes based on biomass and phenology, how is phenology dynamically predicted by the model?

Reply: **Yes, aDGM2 predict change in phenology. In Scheiter et al., (2020) we did a detailed analysis of phenology changes from deciduous to evergreen types that can be explained by increasing precipitation and reduced transpirational demand in $C_3$ plants.**

**Here is brief summary of phenological model implemented in aDGVM2:**

**The aDGVM2 simulates four different phenological strategies: light-triggered evergreen, rain-triggered evergreen, light-triggered deciduous, and rain-triggered deciduous (Langan et al. 2017). Woody plants can adopt all four types whereas we assume that grasses are evergreen. Whether a plant is deciduous or evergreen and whether it is light- or water-triggered are two dynamic traits that are constant during the lifespan of a plant, but that can change between generations due to trait inheritance and community assembly processes in aDGVM2. Deciduous vegetation switches between a dormant and a metabolically active state once moving averages of soil matric potential (water-triggered) or solar radiation (light-triggered) exceed or fall below threshold values. Evergreen woody plants remain metabolically active during their entire life time. However, leaf flushing of evergreen plants is stimulated by water and light triggers, i.e., leaf flushing occurs once moving averages of soil matric potential (water-triggered) or solar radiation (light-triggered) exceed threshold values. The threshold values are plant-specific dynamic traits. They are constant during the lifespan of a plant but they can change between generations due to trait inheritance and the community assembly processes in aDGVM2. While our**

**approach does not allow plants to switch between phenological strategies during their lifetime, growing season length can adjust to inter-annual variation of the climate, because the thresholds used to trigger phenology can be crossed earlier or later in the year.**

**In asGVM2, simulated phenology is considered in the classification scheme and biome changes related to phenology are caused by changes in the abundance/biomass/cover of these different phenological strategies.**

4) Biodiversity and conservation were mentioned throughout the manuscript. But these are implications, not directly addressed in this current work. I would suggest authors to focus on SPECIFIC contributions of the current work and avoid making overstatements.

**Reply: The findings of our current study have specific implication for ecosystem management. For example, woody encroachment predicted in the open savanna biomes implies change in the current biome state that would threatens the biodiversity of the system and affect the wildlife. Another example is that woody encroachment in arid regions would affect water resources in the semi-arid.**

**We have highlighted these major implication of our current find in bold in line 497-500 and 504-506.**

---

## Author Response (AR4)

Dear Dr. Eyal Rotenberg,

Thank you for accepting our manuscript. We would also like to thank you as well as the reviewers for their suggestions and comments which helped us to improve the manuscript.

We corrected the technical issue with the units of evapotranspiration.

Best Regards,
Dushyant Kumar and Co-authors.

**Comments to the Author:**
**To the Authors,**
Following 5 massive comments by different reviewers it is time to wrap this paper.
A technical note to address: if water fluxes are present in mm the units are mm/time, not mm/m2/time. The comment applies to lines 241-2 in the manuscript.
I would like to thank you for the determination and for the larger efforts, good luck with the submission, good health and happy holiday time, Eyal

Reply: **Thank you for pointing it out. We double checked and corrected the units of the evapotranspiration to mm/year.**